# SCHEMA: STATE CHANGES MATTER FOR PROCEDURE PLANNING IN INSTRUCTIONAL VIDEOS

**Yulei Niu    Wenliang Guo**[1]    **Long Chen**[2]    **Xudong Lin**[1]    **Shih-Fu Chang**[1]
[1]Columbia University    [2]The Hong Kong University of Science and Technology
`yn.yuleiniu@gmail.com`

## ABSTRACT

We study the problem of procedure planning in instructional videos, which aims to make a goal-oriented sequence of action steps given partial visual state observations. The motivation of this problem is to learn a *structured and plannable state and action space*. Recent works succeeded in sequence modeling of *steps* with only sequence-level annotations accessible during training, which overlooked the roles of *states* in the procedures. In this work, we point out that State CHangEs MAtter (SCHEMA) for procedure planning in instructional videos. We aim to establish a more structured state space by investigating the causal relations between steps and states in procedures. Specifically, we explicitly represent each step as state changes and track the state changes in procedures. For step representation, we leveraged the commonsense knowledge in large language models (LLMs) to describe the state changes of steps via our designed chain-of-thought prompting. For state change tracking, we align visual state observations with language state descriptions via cross-modal contrastive learning, and explicitly model the intermediate states of the procedure using LLM-generated state descriptions. Experiments on CrossTask, COIN, and NIV benchmark datasets demonstrate that our proposed SCHEMA model achieves state-of-the-art performance and obtains explainable visualizations.

## 1    INTRODUCTION

Humans are natural experts in procedure planning, *i.e.*, arranging a sequence of instruction steps to achieve a specific goal. Procedure planning is an essential and fundamental reasoning ability for embodied AI systems and is crucial in complicated real-world problems like robotic navigation (Tellex et al., 2011; Jansen, 2020; Brohan et al., 2022). Instruction steps in procedural tasks are commonly state-modifying actions that induce *state changes* of objects. For example, for the task of "`grilling steak`", a *raw* steak would be first *topped with pepper* after "`seasoning the steak`", then *placed on the grill* before "`closing the lid`", and become *cooked pieces* after "`cutting the steak`". These before-states and after-states reflect fine-grained information like shape, color, size, and location of entities. Therefore, the planning agents need to figure out both the temporal relations between action steps and the causal relations between steps and states.

Instructional videos are natural resources for learning procedural activities from daily tasks. Chang et al. (2020) proposed the problem of procedure planning in instructional videos, which is to produce a sequence of action steps given the visual observations of start and goal states, as shown in Figure 1 (a). The motivation for this problem is to *learn a structured and plannable state and action space* (Chang et al., 2020). While earlier works (Chang et al., 2020; Sun et al., 2022; Bi et al., 2021) utilized the full annotations of step sequences and intermediate states as supervision (Figure 1(b)), recent works (Zhao et al., 2022; Wang et al., 2023b) achieved promising results with weaker supervision, where only step sequence annotations are available during training (Figure 1(c)). The weaker supervision setting reduces the expensive cost of video annotations and verifies the necessity of *plannable action space*. However, as the intermediate visual states are excluded during both training and evaluation, how to comprehensively represent the state information remains an open question.

This paper aims to establish a more *structured state space* for procedure planning by investigating the causal relations between steps and states in procedures. We first ask: how do humans recognize and

---

Code: `https://github.com/WenliangGuo/SCHEMA`

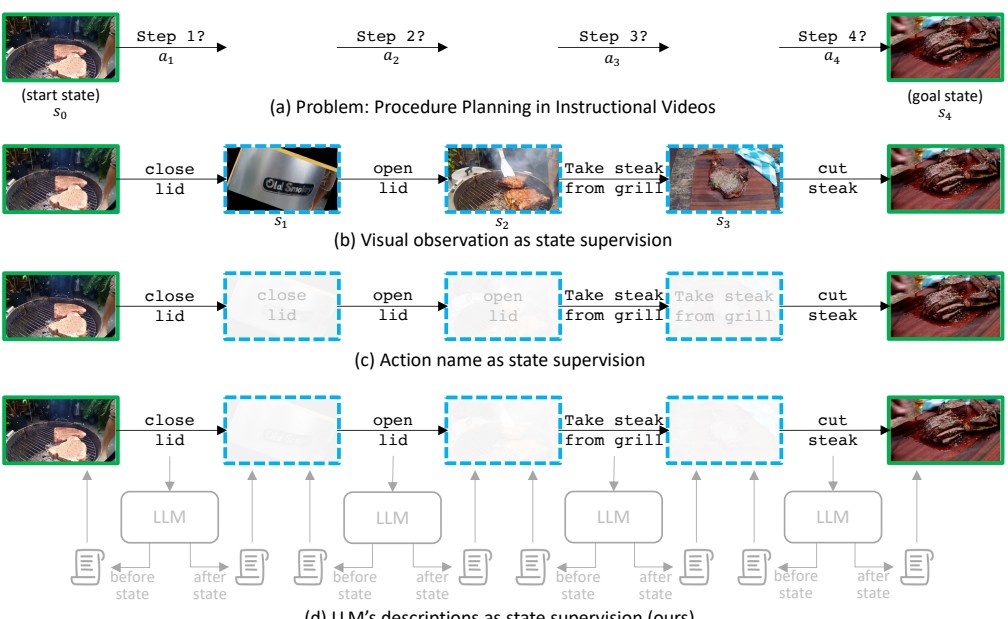

Figure 1: (a) The procedure planning is to predict a sequence of action steps given the visual observations of start and goal states. (b) For the full supervision setting, intermediate states are annotated with timestamps in the videos. (c) For the weak supervision setting, only step names are annotated and taken as supervisions of intermediate states. (d) We represent each step as state changes and take LLMs-generated descriptions for state representation learning.

understand and distinguish steps in procedures? Instead of solely focusing on the action information, humans would track state changes in the procedures by leveraging their commonsense knowledge, which is more informative than only looking at the actions. For example, the steak is *cooked and as a whole* before "cutting" for "grilling steak", and becomes *cooked pieces* after the step. Previous NLP studies have demonstrated the helpfulness of state change modeling in various reasoning tasks, including automatic execution of biology experiments (Mysore et al., 2019), cooking recipes (Bollini et al., 2013), and daily activities (Yang & Nyberg, 2015). Recent studies further explicitly track state changes of entities in procedure texts (Mishra et al., 2018; Tandon et al., 2018; 2020; Zhang et al., 2023; Wu et al., 2023; Li & Huang, 2023). The success of state changes modeling motivates us to investigate the causal relations between steps and states for procedure planning.

In this work, we achieve this goal by representing each state-modifying step as state changes. The cores of our method are step representation and state change tracking. For step representation, motivated by the success of large language models (LLMs) in visual recognition (Menon & Vondrick, 2022), we leveraged LLMs to describe the state changes of each step. Specifically, we asked LLMs (e.g., GPT-3.5) to describe the states before and after each step with our designed chain-of-thought prompts (Sec. 3.2). For state changes tracking, as shown in Figure 1(d), we align the visual state observations with language state descriptions via cross-modal contrastive learning. Intuitively, the start visual state should be aligned with the before-state descriptions of the first step, while the goal visual state should be aligned with the after-state descriptions of the last step. As the language descriptions are more discriminative than visual states, we expect the multi-modal alignment to learn a more structured state space. We also take state descriptions as supervisions of intermediate visual states. Finally, the step prediction model is trained in a masked token prediction manner.

Our main contributions are summarized as follows:

- We pointed out that State CHangEs MAtter (SCHEMA) for procedure planning in instructional videos, and proposed a new representation of steps in procedural videos as state changes.

- We proposed to track state changes by aligning visual state observations with LLMs-generated language descriptions for a more structured state space and represent mid-states via descriptions.

- Our extensive experiments on CrossTask, COIN, and NIV datasets demonstrated the quality of state description generation and the effectiveness of our method.

## 2  RELATED WORK

**Procedure Planning** (Chang et al., 2020; Zhang et al., 2020) is an essential and fundamental problem for embodied agents. In this work, we followed Chang *et al.*'s (Chang et al., 2020) formulation of procedure planning. Recent works proposed different approaches for sequence generation, *e.g.*, autoregressive Transformers (Sun et al., 2022), policy learning (Bi et al., 2021), probabilistic modeling (Bi et al., 2021), and diffusion models (Wang et al., 2023b). Interestingly, Zhao et al. (2022) used only language instructions as supervision for procedures and did not require full annotations of intermediate visual states, which highlights the importance of sequence generation for procedure planning. These methods commonly formulated the problem of procedure planning as conditional sequence generation, and the visual observations of states are treated as conditional inputs. However, the motivation of procedure planning is to align the state-modifying actions with their associated state changes, and expect the agents to understand how the state changes given the actions.

**Instructional Videos Analysis**. Instructional videos have been a good data source to obtain data for procedural activities (Rohrbach et al., 2016; Kuehne et al., 2014; Zhou et al., 2018; Zhukov et al., 2019; Tang et al., 2019; Miech et al., 2019). Existing research on this topic usually tackles understanding the step-task structures in instructional videos, where the step/task annotation can be either obtained from manual annotation on a relatively small set of videos (Zhou et al., 2018; Zhukov et al., 2019; Tang et al., 2019) or through weak/distant supervision on large unlabeled video data (Miech et al., 2019; 2020; Dvornik et al., 2023; Lin et al., 2022). For example, StepFormer (Dvornik et al., 2023) is a transformer decoder trained with video subtitles (mostly from automatic speech recognition) for discovering and localizing steps in instructional videos. Another recent research (Souček et al., 2022a) tackles a more fine-grained understanding of instructional videos, which learns to identify state-modifying actions via self-supervision. However, such approaches require training on the large collection of noisy unlabeled videos, which is expensive and inaccurate enough (Souček et al., 2022a).

**Tracking State Changes** is an essential reasoning ability in complex tasks like question answering and planning. Recent work has made continuous progress on explicitly tracking entity state changes in procedural texts (Mishra et al., 2018; Tandon et al., 2018; 2020; Zhang et al., 2023; Wu et al., 2023; Li & Huang, 2023). Some work in the CV area also investigated the relations between actions and states in videos (Alayrac et al., 2017; Souček et al., 2022a;b; Nishimura et al., 2021; Shirai et al., 2022; Xue et al., 2023; Souček et al., 2023; Saini et al., 2023). Especially, Nishimura et al. (2021) focused on the video procedural captioning task and proposed to model material state transition from visual observation, which introduces a visual simulator modified from a natural language understanding simulator. Shirai et al. (2022) established a multimodal dataset for object state change prediction, which consists of image pairs as state changes and workflow of receipt text as an action graph. However, the object category is limited to food or tools for cooking. Considering the similarity between procedural texts and instructional videos it is natural to explore state changes in instructional videos. In this work, we investigate state changes in procedural videos for procedure planning.

## 3  SCHEMA: STATE CHANGES MATTER

In this section, we introduce the details of our proposed framework, State CHangEs MAtter (SCHEMA). We first introduce the background of procedure planning in Sec. 3.1, and present our method in Sec. 3.2~3.4. Specifically, we first provide the details of step representation in Sec. 3.2, model architecture in Sec. 3.3, and training and inference in Sec. 3.4.

### 3.1  PROBLEM FORMULATION

We follow Chang et al. (2020)'s formulation of procedure planning in instructional videos. As shown in Figure 1(a), given the visual observations of start state $s_0$ and goal state $s_T$, the task is to plan a procedure, *i.e.*, a sequence of action steps $\hat{\pi} = a_{1:T}$, which can transform the state from $s_0$ to $s_T$. The procedure planning problem can be formulated as $p(a_{1:T}|s_0, s_T)$.

The motivation of this task is to learn a *structured and plannable state and action space*. For the training supervision, earlier works used full annotations of procedures including both action steps $a_{1:T}$ and their associated visual states, *i.e.*, the states before and after the step, which are annotated as timestamps of videos (Figure 1(b)). Zhao et al. (2022) proposed to use weaker supervision

```
[goal]: Make Kimchi Fried Rice            [goal]: Make Pancakes
[step]: add onion                         [step]: pour milk
Step Description:                         Step Description:
- Add diced onion into the fried rice.    - Pour milk into the pancake batter.
Before:                                   Before:
- The diced onion is separate from the pan. - The milk is in a container.
- The pan contains fried rice.            - The pancake batter contains no milk.
- The pan has no onion on it.             - The milk is a liquid.
After:                                    After:
- The diced onion is mixed with the fried rice. - The milk is mixed with the pancake batter.
- The onion is on the pan.                - The milk is in the mixing bowl.
- The pan contains onion.                 - The pancake batter contains milk.
```

Figure 2: Examples of GPT-3.5 generated descriptions using our chain-of-thought prompting.

where only the action step annotations $a_{1:T}$ are available (Figure 1(c)), which reduces the expensive annotation cost for videos. Recent works under this setting show that the *plannable action space* can be established by conditional sequence generation (Zhao et al., 2022; Wang et al., 2023b). However, there remain open questions about the role of *structured state space*: why are intermediate states optional for procedure planning? Are there any better representations for steps and visual states?

## 3.2 STEP REPRESENTATION AS STATE CHANGES IN LANGUAGE

Our goal is to construct a more *structured state space* by investigating the causal relations between steps and states in procedures. Motivated by the success of state changes modeling in various reasoning tasks (Bollini et al., 2013; Yang & Nyberg, 2015; Mysore et al., 2019), we represent steps as their before-states and after-states. The state changes can be represented by visual observations or language descriptions. We observed that visual scenes in instructional videos are diverse and noisy, and the details are hard to capture if the object is far from the camera. In addition, the intermediate visual states may not be available due to the high cost of video annotations. Therefore, we represent state changes as discriminative and discrete language descriptions.

Motivated by Menon & Vondrick (2022), we leveraged large language models (LLMs), such as GPT-3.5 (Brown et al., 2020), to generate language descriptions of states based on their commonsense knowledge. In short, we fed each action step with its high-level task goal to the LLMs, and query several descriptions about the associate states before and after the action step. A baseline prompting following Menon & Vondrick (2022) for state descriptions is:

```
Q: What are useful features for distinguishing the states
   before and after [step] for [goal] in a frame?
A: There are several useful visual features to tell the state
   changes before and after [step] for [goal]:
```

However, we empirically found that this prompting does not work well for state descriptions, as LLMs disregard the commonsense knowledge behind the step. For example, given the action step "add onion" and the task "make kimchi fried rice", the before-state description is "the onion was uncut and unchopped", which is incorrect because the onion should have been cut.

**Chain-of-thought Prompting**. Aware of the misalignment between action and states, we proposed a chain-of-thought prompting (Wei et al., 2022) strategy to first describe the details of action steps and then describe the state changes according to the details of the steps. Our prompt is designed as:

```
First, describe the details of [step] for [goal] with one verb.
Second, use 3 sentences to describe the status changes of
   objects before and after [step], avoiding using [verb].
```

where "[verb]" is the action name (*e.g.*, "pour") to increase the description diversity. We also provide several examples as context (see appendix). We fixed the number of descriptions as 3 as we empirically found that one or two descriptions cannot cover all the objects and attributes, while more than three descriptions are redundant. Figure 2 illustrates two examples of the generated descriptions based on our prompts. Overall, the step and state descriptions contain more details about attributes, locations, and relations of objects. In the following, for the step name $A_i$, we denote its step description as $d_i^s$, before-state descriptions as $\{d_{i1}^b, \cdots, d_{iK}^b\}$, and after-state descriptions as $\{d_{i1}^a, \cdots, d_{iK}^a\}$, where $K=3$ in our implementation.

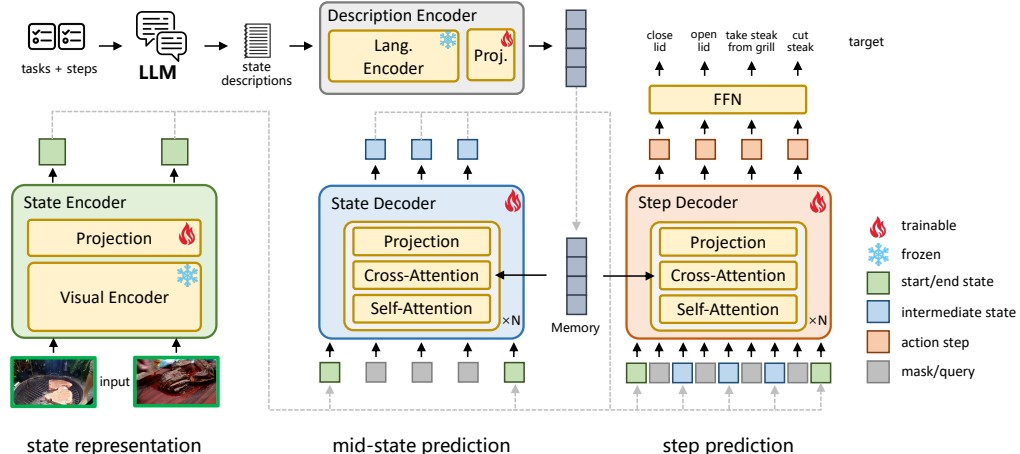

Figure 3: The pipeline of our SCHEMA for procedure planning.

### 3.3 ARCHITECTURE

Figure 3 illustrates the overview of our SCHEMA pipeline. Overall, we break up the procedure planning problem $p(a_{1:T}|s_0, s_T)$ into two subproblems, *i.e.*, mid-state prediction and step prediction. Mid-state prediction is to estimate the intermediate states $s_{1:(T-1)}$ given $s_0$ and $s_T$, *i.e.*, $p(s_{1:(T-1)}|s_0, s_T)$. Step prediction is to predict the step sequence given the full states, *i.e.*, $p(a_{1:T}|s_{0:T})$. We formulate the procedure planning problem as:

$$p(a_{1:T}|s_0, s_T) = \int \underbrace{p(a_{1:T}|s_{0:T})}_{\text{step prediction}} \underbrace{p(s_{1:(T-1)}|s_0, s_T)}_{\text{mid-state prediction}} ds_{1:(T-1)}. \tag{1}$$

#### 3.3.1 STATE REPRESENTATION

We align visual observations with language descriptions of the same states to learn a structure state space, which will be introduced in Sec. 3.4.

**State encoder**. The state encoder takes the video frame as input and outputs its embedding. The state encoder $f_s^{enc}$ consists of a fixed pre-trained visual feature extractor $f^{vis}$ and a trainable projection (two-layer FFN) $f_s^{proj}$. The embedding for state $s$ is obtained by $s^{enc} = f_s^{enc}(s) = f_s^{proj}(f^{vis}(s))$.

**Description encoder**. Similar to the state encoder, the description encoder $f_d^{enc}$ consists of a fixed language feature extractor $f^{lang}$ and a trainable projection $f_d^{proj}$. The description encoder takes description $d$ as input and outputs its embedding $d^{enc} = f_d^{enc}(d) = f_d^{proj}(f^{lang}(d))$.

#### 3.3.2 MID-STATE PREDICTION

**State Decoder**. The state decoder $f_s^{dec}$ is an non-autoregressive Transformer (Vaswani et al., 2017). The state decoder predicts the intermediate states $s_{1:(T-1)}$ given the start state $s_0$ and the goal state $s_T$. The query for the state decoder is denoted as $Q_s = [s_0^{enc} + p_0, p_1, \cdots, p_{T-1}, s_T^{enc} + p_T]$, where $p_i$ denotes the $i$-th positional embedding. The state decoder also takes the collection of state descriptions $D_s = \{d_{11}^b, \cdots, d_{CK}^b, d_{11}^a, \cdots, d_{CK}^a\}$ as the external memory $M$, where $C$ is the number of step classes and $M = f_d^{enc}(D_s)$. The external memory interacts with the decoder via cross-attention. The state decoding process to obtain the embeddings $\hat{s}_i^{dec}$ is denoted as:

$$\hat{s}_1^{dec}, \cdots, \hat{s}_{T-1}^{dec} = f_s^{dec}(Q_s, M). \tag{2}$$

#### 3.3.3 STEP PREDICTION

**Step Decoder**. The step decoder $f_a^{dec}$ is a Transformer model with a similar architecture as the state decoder $f_s^{dec}$. The query combines state embeddings and positional embeddings, denoted as $Q_a = [s_0^{enc} + q_0, q_1, \hat{s}_1^{dec} + p_2, \cdots, \hat{s}_{T-1}^{dec} + q_{2T-2}, q_{2T-1}, s_T^{enc} + q_{2T}]$ where $q_i$ denotes the

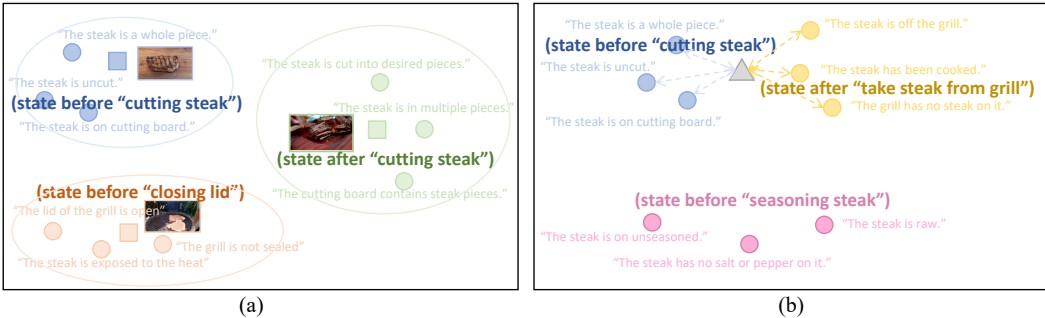

Figure 4: (a) For structured state space learning, we force the visual state observation to be close to its language descriptions and away from others. (b) For mid-state prediction, we use the before-state descriptions and after-state descriptions as guidance to learn the presentation of mid-states.

$i$-th positional embedding. Similar to the state decoder $f_s^{dec}$, the step decoder $f_a^{dec}$ also takes $M = f_d^{enc}(D_s)$ as the external memory. The step decoding process is denoted as:

$$\hat{a}_1^{dec}, \cdots, \hat{a}_T^{dec} = f_a^{dec}(Q_a, M), \tag{3}$$

where $\hat{a}_1^{dec}, \cdots, \hat{a}_T^{dec}$ are the estimated action embeddings. A two-layer feed-forward network (FFN) $f_a^{cls}$ is built on top of $\hat{a}^{dec}$ as the step classifier to predict the logits of steps, *i.e.*, $\hat{a} = f_a^{cls}(\hat{a}^{dec})$.

In addition to capturing the task information, we establish a task classifier that takes the visual features of start and end states as input and outputs a vector to represent the task information (Wang et al., 2023b;a). The task feature is added to the queries $Q_s$ and $Q_a$, where we omitted it for simplicity.

## 3.4 TRAINING AND INFERENCE

The training process consists of three parts: (1) state space learning that aligns visual observations with language descriptions, (2) masked state modeling for mid-state prediction, and (3) masked step modeling for step prediction. For simplicity, we define the losses with one procedure.

**State Space Learning**. Although vision-language models like CLIP (Radford et al., 2021) are pre-trained for vision-language alignment, they cannot be directly used for our problem as the pre-training is not designed for fine-grained state understanding. Therefore, we train two additional projections $f_s^{proj}$ and $f_d^{proj}$ on top of the visual encoder and language encoder. The added projections also allow us to align other visual features with language features. Given the start state $s_0$ (or end state $s_T$) and a step label $a_i$, the similarity between $s_0$ (or $s_T$) and each step $A_i$ is calculated by $sim(s_0, A_i) = \sum_{j=1}^K <s_0^{enc}, d_{ij}^{enc,b}>$ and $sim(s_T, A_i) = \sum_{j=1}^K <s_T^{enc}, d_{ij}^{enc,a}>$, where $< \cdot, \cdot >$ denotes the dot product. Figure 4 (a) illustrates the idea of structured state space via vision-language alignment. Specifically, we regard the language descriptions with the same state as positive samples, and take descriptions of the other states as negative samples. We define the contrastive loss as:

$$L_s^{enc} = -\log \underbrace{\frac{\exp(sim(s_0, A_{a_1}))}{\sum_{i=1}^C \exp(sim(s_0, A_i))}}_{\text{start state}} - \log \underbrace{\frac{\exp(sim(s_T, A_{a_T}))}{\sum_{i=1}^C \exp(sim(s_T, A_i))}}_{\text{end state}} \tag{4}$$

**Masked State Modeling**. The mid-state prediction process can be regarded as a masked state modeling problem, where the intermediate states are masked from the state sequence, and the state decoder recovers the masked states. Since the annotations of intermediate states in videos are not available, we instead use LLMs-generated state descriptions as guidance. In a procedure $(s_0, a_1, s_1, \cdots, s_{T-1}, a_T, s_T)$ where the mid-state $s_t$ is the after-state for action $a_t$ and before-state for action $a_{t+1}$, we use the before-state descriptions of $a_t$ and after-state descriptions of $a_{t+1}$ as the supervision for $s_t$. Figure 4(b) illustrates the idea of mid-state learning. We average the description embeddings as the target embedding $\hat{s}_t^{dec}$ for $s_t$, and calculate the mean squared error between $\hat{s}_t^{dec}$

Table 1: Comparison with other methods on CrossTask dataset.

| Models | T = 3 | | | T = 4 | | | T=5 | T = 6 |
|---|---|---|---|---|---|---|---|---|
| | SR↑ | mAcc↑ | mIoU↑ | SR↑ | mAcc↑ | mIoU↑ | SR↑ | SR↑ |
| Random | <0.01 | 0.94 | 1.66 | <0.01 | 0.83 | 1.66 | <0.01 | <0.01 |
| Retrieval-Based | 8.05 | 23.30 | 32.06 | 3.95 | 22.22 | 36.97 | 2.40 | 1.10 |
| WLTDO | 1.87 | 21.64 | 31.70 | 0.77 | 17.92 | 26.43 | — | — |
| UAAA | 2.15 | 20.21 | 30.87 | 0.98 | 19.86 | 27.09 | — | — |
| UPN | 2.89 | 24.39 | 31.56 | 1.19 | 21.59 | 27.85 | — | — |
| DDN | 12.18 | 31.29 | 47.48 | 5.97 | 27.10 | 48.46 | 3.10 | 1.20 |
| PlaTe | 16.00 | 36.17 | 65.91 | 14.00 | 35.29 | 55.36 | — | — |
| Ext-MGAIL w/o Aug. | 18.01 | 43.86 | 57.16 | — | — | — | — | — |
| Ext-GAIL | 21.27 | 49.46 | 61.70 | 16.41 | 43.05 | 60.93 | — | — |
| P³IV w/o Adv. | 22.12 | 45.57 | 67.40 | — | — | — | — | — |
| P³IV | 23.34 | 49.96 | 73.89 | 13.40 | 44.16 | 70.01 | 7.21 | 4.40 |
| EGPP | 26.40 | 53.02 | 74.05 | 16.49 | 48.00 | 70.16 | 8.96 | 5.76 |
| SCHEMA (Ours) | **31.83** | **57.31** | **78.33** | **19.71** | **51.85** | **74.46** | **11.41** | **7.68** |

and $s_t^{dec}$ as the mid-state prediction loss:

$$s_t^{dec} = \frac{1}{2K}(\sum_{j=1}^{K} d_{a_{t,j}}^{enc,a} + d_{a_{t+1,j}}^{enc,b}), \qquad L_s^{dec} = \sum_{t=1}^{T-1}(\hat{s}_t^{dec} - s_t^{dec})^2. \qquad (5)$$

**Masked Step Modeling**. Similar to mid-state estimation, the step prediction process can also be regarded as a masked step modeling problem, where the steps are masked from the state-action sequences. The loss is the cross-entropy between ground-truth answers $a_t$ and predictions $\hat{a}_t^{dec}$, *i.e.*, $L_a^{dec} = \sum_{t=1}^{T} -a_t \log \hat{a}_t^{dec}$. The final loss combines the above losses, *i.e.*, $L = L_s^{enc} + L_s^{dec} + L_a^{dec}$.

**Inference**. The non-autoregressive Transformer model may make insufficient use of the temporal relation information among the action steps, *i.e.*, action co-occurrence frequencies. Inspired by the success of Viterbi algorithm(Viterbi, 1967) in sequential labeling works (Koller et al., 2016; 2017; Richard et al., 2017; 2018), we follow Zhao et al. (2022) and conduct the Viterbi algorithm for post-processing during inference. For Viterbi, we obtained the emission matrix based on the probability distribution over $[\hat{a}_1^{dec}, \cdots, \hat{a}_T^{dec}]$, and estimated the transition matrix based on action co-occurrence frequencies in the training set. Different from Zhao et al. (2022) that applied Viterbi to probabilistic modeling, we applied Viterbi to deterministic modeling. Specifically, instead of sampling 1,500 generated sequences to estimate the emission matrix (Zhao et al., 2022), we run the feedforwarding only once and use the single predicted probability matrix as the emission matrix, which is simpler and more efficient.

## 4 EXPERIMENTS

### 4.1 EVALUATION PROTOCOL

**Datasets**. We evaluate our SCHEMA method on three benchmark instruction video datasets, CrossTask (Zhukov et al., 2019), and COIN (Tang et al., 2019), and NIV (Alayrac et al., 2016). The CrossTask dataset consists of 2,750 videos from 18 tasks depicting 133 actions, with an average of 7.6 actions per video. The COIN dataset contains 11,827 videos from 180 tasks, with an average of 3.6 actions per video. The NIV dataset contains 150 videos with an average of 9.5 actions per video. Following previous works (Chang et al., 2020; Bi et al., 2021; Sun et al., 2022), we randomly select 70% of the videos in each task as the training set and take the others as the test set. We extract all the step sequences $a_{t:(t+T-1)}$ in the videos as procedures with the time horizon $T$ as 3 or 4.

**Feature Extractors**. As our goal for state space learning is to align visual observation and language descriptions, we tried CLIP (Radford et al., 2021) ViT-L/14 model as visual encoder and its associated pretrained Transformer as language encoder. We also follow recent works and use the S3D network (Miech et al., 2019) pretrained on the HowTo100M dataset (Miech et al., 2020) as the visual encoder, and add two projections for vision-language alignment (Sec. 3.4). We empirically found

Table 2: Comparison with PDPP on CrossTask dataset. † denotes the results under PDPP's setting.

| Models | T = 3 | | | T = 4 | | | T=5 | T = 6 |
|---|---|---|---|---|---|---|---|---|
| | SR↑ | mAcc↑ | mIoU↑ | SR↑ | mAcc↑ | mIoU↑ | SR↑ | SR↑ |
| PDPP | 26.38 | 55.62 | 59.34 | 18.69 | **52.44** | 62.38 | **13.22** | 7.60 |
| SCHEMA (Ours) | **31.83** | **57.31** | **78.33** | **19.71** | 51.85 | **74.46** | 11.41 | **7.68** |
| PDPP† | 37.20 | **64.67** | 66.57 | 21.48 | 57.82 | 65.13 | 13.45 | 8.41 |
| SCHEMA (Ours)† | **38.93** | 63.80 | **79.82** | **24.50** | **58.48** | **76.48** | **14.75** | **10.53** |

Table 3: Comparisons with other methods on COIN dataset.

| Models | T = 3 | | | T = 4 | | |
|---|---|---|---|---|---|---|
| | SR↑ | mAcc↑ | mIoU↑ | SR↑ | mAcc↑ | mIoU↑ |
| Random | <0.01 | <0.01 | 2.47 | <0.01 | <0.01 | 2.32 |
| Retrieval | 4.38 | 17.40 | 32.06 | 2.71 | 14.29 | 36.97 |
| DDN | 13.90 | 20.19 | 64.78 | 11.13 | 17.71 | 68.06 |
| P³IV | 15.40 | 21.67 | 76.31 | 11.32 | 18.85 | 70.53 |
| EGPP | 19.57 | 31.42 | **84.95** | 13.59 | 26.72 | **84.72** |
| SCHEMA (Ours) | **32.09** | **49.84** | 83.83 | **22.02** | **45.33** | 83.47 |

that video-based pre-trained features perform better than image-based pre-trained features. In the following, we used the pretrained S3D model as visual encoder.

**Metrics**. Following previous works (Chang et al., 2020; Bi et al., 2021; Sun et al., 2022; Zhao et al., 2022), we evaluate the models on three metrics. (1) Success Rate (SR) is the most strict metric that regards a procedure as a success if all the predicted action steps in the procedure match the ground-truth steps. (2) mean Accuracy (mAcc) calculates the average accuracy of the predicted actions at each step. (3) mean Intersection over Union (mIoU) is the least strict metric that calculates the overlap between the predicted procedure and ground-truth plan, obtained by $\frac{|\{a_t\} \cap \{\hat{a}_t\}|}{|\{a_t\} \cup \{\hat{a}_t\}|}$, where $\{\hat{a}_t\}$ is the set of predicted actions and $\{a_t\}$ is the set of ground truths.

**Baselines**. We follow previous works and consider the following baseline methods for comparisons. The recent baselines are (1) PlaTe (Sun et al., 2022), which extends DNN and uses a Transformer-based architecture; (2) Ext-GAIL (Bi et al., 2021), which uses reinforcement learning for procedure planning; (3) P³IV (Zhao et al., 2022), which is the first to use weak supervision and proposed a generative adversarial framework; (4) PDPP (Wang et al., 2023b), which is a diffusion-based model for sequence distribution modeling; and (5) EGPP (Wang et al., 2023a), which extracts event information for procedure planning. Details of other earlier baselines can be found in appendix.

## 4.2 RESULTS

**Comparisons with Other Methods**. Tables 1 and 3 show the comparisons between our method and others on CrossTask and COIN datasets. Overall, our proposed method outperforms other methods by large margins on all the datasets and metrics. Specifically, for $T = 3$ on CrossTask, our method outperforms P³IV by over 8% (31.83 vs. 23.34) on the sequence-level metric SR and outperforms EGPP by over 5% (31.83 vs. 26.40). The improvements are consistent with longer procedures (*i.e.*, $T = 4, 5, 6$), and other two step-level metrics mAcc and mIoU. We also found that both P³IV and EGPP didn't work well on COIN compared to their performances on CrossTask. Specifically, P³IV outperforms DDN by large margins on CrossTask (*e.g.*, 23.34 vs. 12.18 for SR with $T = 3$). However, the improvements on COIN become marginal, especially for longer procedures (*i.e.*, 11.32 vs. 11.13 for SR with $T = 4$). The similar results are observed on EGPP, As comparisons, the SR of our SEPP is consistently larger than P3IV by over 16% for $T = 3$ and 10% for $T = 4$. The improvements of mACC and mIoU are also significant. These results demonstrate the better generality and effectiveness of our method on different datasets compared to P3IV and EGPP.

An exception case is the recent work PDPP (Wang et al., 2023b) which achieves higher performance on both datasets. However, we argued that they define the start state and end state differently.

Table 4: Ablation studies on state space learning and mid-state prediction on CrossTask with $T=3$.

|  | State Align. | Mid-state Pred. | SR↑ | mAcc↑ | mIoU↑ |
|---|---|---|---|---|---|
| (a) |  |  | 28.72 | 54.72 | 76.66 |
| (b) |  | ✓ | 29.41 | 54.92 | 77.26 |
| (c) | ✓ |  | 30.15 | 56.32 | 77.68 |
| (d) | ✓ | ✓ | **31.83** | **57.31** | **78.33** |

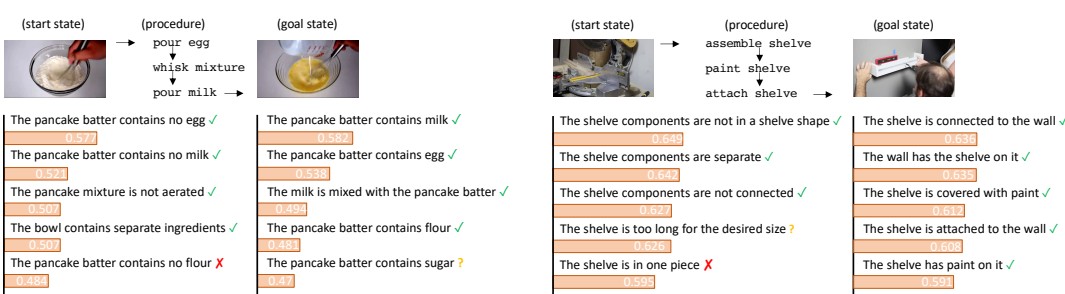

Figure 5: Examples of state justifications from our model.

Specifically, previous works define states as a 2-second window *around* the start/end time, while PDPP defines the window *after* the start time and *before* the end time. Such a definition is more likely to access step information especially for short-term actions, leading to unfair comparisons with other methods. We further compared our method with PDPP under both conventional setting and their setting. The results on Table 2 show that our method outperforms PDPP with $T = 3$ and $T = 4$ under both settings, and the main improvement of PDPP comes from the different setting with a small $T$ (*e.g.*, ∼11% increase of SR on $T=3$). An interesting observation is that the benefits of different settings to PDPP become marginal with a larger $T$, which may be credited to their diffusion model.

**Ablation Studies**. We first conduct ablation studies on CrossTask to validate the effect of two key components, state alignment (Eq. 4 and Figure 4(a)) and mid-state prediction (Sec. 3.3.2). As shown in Table 4, visual-language alignment improves SR by 1.4∼2.4% ((c) vs. (a), (d) vs. (b)) for $T = 3$ on CrossTask. Also, the mid-state prediction module also improves the performance, and the improvements become larger with state alignment (*i.e.*, +0.69% on SR w/o state alignment vs. +1.68% on SR with state alignment). The entire combination (d) achieves the best results. These results verified the impacts of state space learning and mid-state prediction. More ablation studies are in the appendix.

**Qualitative Results**. Figure 5 illustrates examples of state justifications, *i.e.*, how the model aligns visual state observation with language descriptions. We retrieve top-5 similar descriptions from the corpus of state descriptions. Overall, the retrieved descriptions match the image well, and most of the top similar descriptions are aligned with the visual observations, which improved the explainable state understanding. More visualization results are in the appendix.

## 5 CONCLUSION

In this work, we pointed out that State CHangEs MAtter (SCHEMA) for procedure planning in instructional videos, and proposed to represent steps as state changes and track state changes in procedural videos. We leveraged large language models (*i.e.*, GPT-3.5) to generate descriptions of state changes, and align the visual states with language descriptions for a more structured state space. We further decompose the procedure planning into two subproblems, mid-state prediction and step prediction. Extensive experiments further verified that our proposed state representation can promote procedure planning. In the future, potential directions are to establish a benchmark dataset to explicitly track state changes in instructional videos, and to investigate the roles of state changes tracking in other procedural learning tasks like pre-training and future step forecasting. In addition, multimodal procedural planning would be a practical and challenging problem that generates coherent textual plans and visual plans, reflecting the state changes in multiple modalities.

ACKNOWLEDGEMENT

This work is supported by U.S. DARPA KAIROS Program No. FA8750-19-2-1004. The views and conclusions contained herein are those of the authors and should not be interpreted as necessarily representing the official policies, either expressed or implied, of DARPA, or the U.S. Government. The U.S. Government is authorized to reproduce and distribute reprints for governmental purposes notwithstanding any copyright annotation therein.

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

## APPENDIX

## A    IMPLEMENTATION DETAILS

In Section 3.3, we introduced the architecture of our SCHEMA. In the appendix, we further introduce the implementation details of the model, including model architecture and training.

**Chain-of-Thought Prompting.** Recall that we use a chain-of-thought prompt with examples to trigger the language model, which encourages the language model to generate the descriptions of state changes according to the action description. For example, for the action step "cut banana" in the task "make banana ice cream", the form of our prompt is:

```
First, describe the details of [step] for [goal] with one verb.
Second, use 3 sentences to describe the status changes of
    objects before and after [step], avoiding using [verb].

[goal]: Grill steak
[step]: season steak
[verb]: season
Description:
Season steak with salt and pepper
Before:
- The steak is unseasoned.
- The steak has no salt or pepper on it.
- The steak is raw.
After:
- The steak is with salt and pepper.
- The steak has salt and pepper on it.
- The steak is ready to be grilled.

[goal]: Make Kimchi Fried Rice
[step]: add ham
[verb]: add
Description:
Incorporate diced ham into the fried rice
Before:
- The diced ham is separate from the pan.
- The pan contains fried rice.
- The pan has no ham on it.
```

```
After:
- The diced ham is blended with the fried rice.
- The ham is on the pan.
- The pan contains ham.

[goal]: Make Banana Ice Cream
[step]: cut banana
```

Note that the examples are generated by GPT-3.5 and manually selected. An example output is

```
[verb]: cut
Description:
Cut banana into small pieces
Before:
- The banana is a whole.
- The banana is uncut.
- The banana is in one piece.
After:
- The banana is in small pieces.
- The banana is cut into pieces.
- The banana is divided into multiple parts.
```

**Transformer Models for State Decoder and Step Decoder.** Our state decoder and step decoder are Transformer-based models. The model consists of two blocks. Each block consists of one self-attention module, one cross-attention module, and a two-layer projection module. The input query is first processed by the self-attention module, then forwarded to the cross-attention module, and processed by the projection module. The cross-attention module takes the external memory to calculate the keys and values. Each self-attention and cross-attention module consists of 32 heads and the hidden layer size is set as 128. The step classifier is a two-layer MLP with hidden size of 128. The dropout ratio is 0.2.

**Training Details.** We train our model with Adam optimizer, an initial learning rate set to $5 \times 10^{-3}$ decayed by 0.65 every 40 epochs. The batch size is set as 256. The training process takes 1 hour (500 epochs) on CrossTask and 5.5 hours (400 epochs) on COIN using a single V100 GPU. We will release the code after the paper is accepted. The code will be released under Apache-2.0 license.

**Viterbi Algorithm for Inference.** During inference, we follow Zhao *et al.* (Zhao et al., 2022) and use Viterbi (Viterbi, 1967) algorithm to further incorporate the temporal prior into the model. Specifically, Viterbi algorithm is conducted based on a transition matrix and an emission matrix. The transition matrix depicts the probability of state transiting from one state to another state, which is denoted as $A \in \mathbb{R}^{N_a \times N_a}$. We use the action co-occurrence frequencies (*i.e.*, ground-truth procedure annotations in the training set) to obtain the transition matrix as temporal prior knowledge. The emission matrix depicts the probability of each state given observations, which is denoted as $B \in \mathbb{R}^{T \times N_a}$ and obtained by the predicted probability distribution.

## B EXPERIMENTS

**Baselines**. We follow previous works (Zhao et al., 2022; Bi et al., 2021; Chang et al., 2020; Sun et al., 2022) and consider the baseline methods for comparisons. In addition to the baselines in Sec 4.1, we introduce other baselines as follows:(1) Random. This baseline randomly selects actions from the candidate action set to generate the procedure. (2) Retrieval-based. The baseline retrieves the closest neighbor of the start and goal

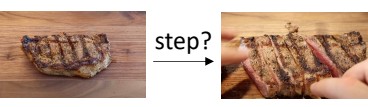

Figure 6: Illustration of state-based step classification.

observations based on the visual feature similarity in the training set. (3) WLTDO (Ehsani et al., 2018). This baseline used a recurrent neural network (RNN) to predict the action sequence given the visual observations. (4) UAAA (Abu Farha & Gall, 2019). This baseline is an auto-regression model using RNN-HMM architecture. (5) UPN (Srinivas et al., 2018). This baseline is a physical-world path planning method. (6) DDN (Chang et al., 2020). This baseline is an auto-regressive model that predicts both the action and state in the same space.

Table 5: Results on the state-based zero-shot step classification task on CrossTask.

| Prompts | None | "A photo of" | "A video of" | "The start/end state of" |
|---|---|---|---|---|
| Acc (%) | 14.6 | 14.1 | 16.0 | 13.4 |

| Descriptions | DCLIP | Action descriptions | State descriptions |
|---|---|---|---|
| Acc (%) | 16.2 | 17.7 | **21.1** |

Table 6: Comparisons on the NIV dataset.

| | $T = 3$ | | | $T = 4$ | | |
|---|---|---|---|---|---|---|
| Models | SR↑ | mAcc↑ | mIoU↑ | SR↑ | mAcc↑ | mIoU↑ |
| Random | 2.21 | 4.07 | 6.09 | 1.12 | 2.73 | 5.84 |
| DDN | 18.41 | 32.54 | 56.56 | 15.97 | 27.09 | 53.84 |
| Ext-GAIL | 22.11 | 42.20 | 65.93 | 19.91 | 36.31 | 53.84 |
| P$^3$IV | 24.68 | 49.01 | 74.29 | 20.14 | 38.36 | 67.29 |
| EGPP | 26.05 | **51.24** | 75.81 | 21.37 | **41.96** | 74.90 |
| SCHEMA (Ours) | **27.93** | 41.64 | **76.77** | **23.26** | 39.93 | **76.75** |

**State Descriptions**. We establish a preliminary task called state-based step classification. As illustrated in Figure 6, given two visual state observations, the task is to predict the one action step that leads to the state changes. It can be regarded as a special form of procedure planning with $T = 1$. We conduct evaluations under the zero-shot setting using CLIP (Radford et al., 2021) with different language descriptions, which are listed in Table 5. We evaluate both manually-designed prompted captions (first row) and GPT-3.5 generated descriptions introduced in Sec. 3.2 (second row). Detailed examples can be found in the appendix. As shown in Table 5, our state descriptions using chain-of-thought prompting achieves a relative 31.9% improvement compared to manually-designed prompted captions (21.1 vs. 16.0), while the improvement of the modified baseline DCLIP (Menon & Vondrick, 2022) is marginal (16.2 vs. 16.0). These results show that our state descriptions are better state descriptors for CLIP.

**Results on NIV**. Table 6 shows the results on the NIV dataset. Similar to the observations on CrossTask and COIN, our SCHEMA achieves better SR and mIoU and competitive mAcc.

**Uncertain Modeling**. Uncertain modeling is to produce several procedures by running the model multiple times with different noise input vectors. Although we focus on deterministic modeling, our model can be easily extended to probabilistic modeling by including the noise vectors into queries of state decoder and step decoder. We follow Zhao et al. (2022) to evaluate the performance of uncertain modeling. Evaluation metrics are KL-Div, NLL, Mode Recall (ModeRec), Mode Precision (ModePre). Results are provided in Table 7. The results of SR, mAcc, and mIoU are shown in Table 8. As shown in the table, the probabilistic variant underperforms the deterministic variant on procedure planning metrics SR, mAcc, and mIoU. The possible reason is that language descriptions as the supervision of state representations would also decreases the uncertainty and variances of visual observations, which conflicts with noisy vectors in the probabilistic variant that increases the uncertainty and variances.

**Ablations on two-decoder design**. We used two decoders for state prediction and step prediction. An alternative is using one decoder for both the state and action steps. We used the two-decoder design because the intermediate states serving as explicit inputs work better than implicit outputs. Table 9 further compares the two designs, showing the effectiveness of the two-decoder design.

**Ablations on external memory**. We further conduct ablation studies to verify the impact of external memory. The baseline is to learn a memory module with the same size of state descriptions but randomly initialized. Table 10 compares different choices of external memory for the Transformer models. As shown in the table, state descriptions works better than random initialized memory, which indicates that the semantic information in state descriptions help with procedure planning.

Note that we also use state descriptions rather than setp descirptions as external memory for the step decoder. Table 11 shows the comparison between these two variants. As shown in the table, state descriptions as external memory outperforms the variant with step descriptions under all the

Table 7: The results of uncertain modeling on the CrossTask dataset.

| Metric | Method | $T = 3$ | $T = 4$ | $T = 5$ | $T = 6$ |
|--------|--------|---------|---------|---------|---------|
| KL-Div ↓ | Ours - determinstic | 4.03 | 4.31 | 4.49 | 4.65 |
|          | Ours - probabilistic | **3.62** | **3.82** | **3.92** | **3.97** |
| NLL ↓ | Ours - determinstic | 4.55 | 5.11 | 5.46 | 5.71 |
|       | Ours - probabilistic | **4.15** | **4.62** | **4.88** | **5.04** |
| ModePrec ↑ | Ours - determinstic | **38.41** | **26.67** | **15.28** | 9.84 |
|            | Ours - probabilistic | 38.32 | 26.46 | 14.99 | **9.91** |
| ModeRec ↑ | Ours - determinstic | 25.59 | 13.63 | 6.27 | 3.37 |
|           | Ours - probabilistic | **37.70** | **23.76** | **13.85** | **9.30** |

Table 8: The planning results of probabilistic model on CrossTask.

| white Models | $T = 3$ | | | $T = 4$ | | | $T=5$ | $T = 6$ |
|--------------|------|-------|-------|------|-------|-------|------|------|
| | SR↑ | mAcc↑ | mIoU↑ | SR↑ | mAcc↑ | mIoU↑ | SR↑ | SR↑ |
| Ours - probabilistic | 29.51 | 57.09 | 77.76 | 16.55 | **51.93** | 74.42 | 8.73 | 5.53 |
| Ours - deterministic | **31.83** | **57.31** | **78.33** | **19.71** | 51.85 | **74.46** | **11.41** | **7.68** |

Table 9: Ablation studies on decoder design.

| | $T = 3$ | | | $T = 4$ | | |
|--|------|-------|-------|------|-------|-------|
| | SR↑ | mAcc↑ | mIoU↑ | SR↑ | mAcc↑ | mIoU↑ |
| One-decoder | 30.25 | 56.35 | 77.83 | 19.01 | 51.15 | 74.13 |
| Two-decoder | **31.83** | **57.31** | **78.33** | **19.71** | **51.85** | **74.46** |

scenarios. The possible reason is that state descriptions contain more information of object status and serve as supplement to step label supervisions. In addition, as shown in Table 5, state descriptions works better than step descriptions for the state-based step classification problem, which indicates that state descriptions are good resources for step recognition.

**Ablations on Viterbi**. We applied Viterbi algorithm to add temporal prior. Table 12 shows that Viterbi has a positive impact on all the metrics, which indicates that Viterbi successfully includes the temporal ordering knowledge in the training data, i.e., action co-occurrence frequencies.

**Qualitative Results**. Figure 7 shows more examples of state justifzication results outputted by our model. These examples show that our model can provide evidence about the visual state, which is more explainable and informative.

**Failure Case Analysis.** The examples are shown in Figure 8. We grouped them into three cases:

(1) Failed understanding of start/end state. As shown in Figure 8(a), the model predicted "season steak" as the first step because it didn't recognize that there is pepper on top of the steak, *i.e.*, it failed to understand the start state. As shown in Figure 8(b), the model predicted "flip steak" rather than "put steak on grill" as the second step. The way to distinguish these two steps is whether the steak has its top side grilled. Although the end state shows that the top side of the steak is raw, the steak is very small to be captured. One future solution is to use high-resolution video frames or object detector to ground the object.

(2) Hallucination. As shown in Figure 8(c), the model predicts "add strawberries to cake" as the third step. However, the goal is not to make strawberry cake. This failure may be due to the training priors as there are many videos for the task of "make french strawberry cake".

(3) Reasonable but not matched with ground-truth plans. As shown in Figure 8(d), the generated plan is reasonable, although it doesn't exactly match the ground-truth annotation. This "failure" indicates that this task needs a better evaluation protocol for all the reasonable results, which is a general challenge for sequence generation tasks.

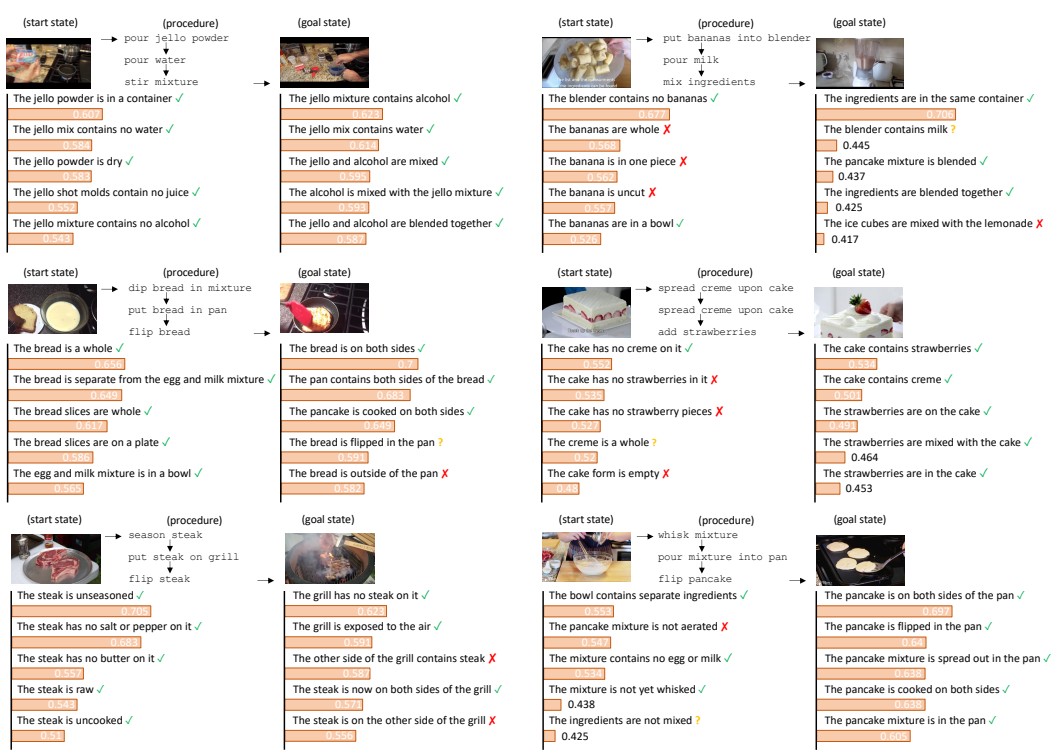

Figure 7: More examples of state justifications from our model.

```
[goal]: Make Kimchi Fried Rice
[step]: add onion
Step Description:
- Add diced onion into the fried rice.
Before:
- The diced onion is separate from the pan.
- The pan contains fried rice.
- The pan has no onion on it.
After:
- The diced onion is mixed with the fried rice.
- The onion is on the pan.
- The pan contains onion.
```

```
[goal]: Make Pancakes
[step]: pour milk
Step Description:
- Pour milk into the pancake batter.
Before:
- The milk is in a container.
- The pancake batter contains no milk.
- The milk is a liquid.
After:
- The milk is mixed with the pancake batter.
- The milk is in the mixing bowl.
- The pancake batter contains milk.
```

Figure 8: Failure case analysis. For each example, the first row is the ground-truth annotation while the second is the prediction.

Table 10: Ablation studies on external memory.

| Memory | $T = 3$ | | | $T = 4$ | | |
|---|---|---|---|---|---|---|
| | SR↑ | mAcc↑ | mIoU↑ | SR↑ | mAcc↑ | mIoU↑ |
| Random Initialized | 29.83 | 56.04 | 77.22 | 19.53 | 50.99 | 73.96 |
| Two-decoder | **31.83** | **57.31** | **78.33** | **19.71** | **51.85** | **74.46** |

Table 11: Ablation studies on memory of step classifier.

| Memory | $T = 3$ | | | $T = 4$ | | | $T = 5$ | $T = 6$ |
|---|---|---|---|---|---|---|---|---|
| | SR↑ | mAcc↑ | mIoU↑ | SR↑ | mAcc↑ | mIoU↑ | SR↑ | SR↑ |
| Step Descriptions | 30.00 | 55.89 | 77.61 | 19.30 | 51.43 | 74.13 | 10.99 | 7.60 |
| State Descriptions | **31.83** | **57.31** | **78.33** | **19.71** | **51.85** | **74.46** | **11.41** | **7.68** |

Table 12: Ablation studies on Viterbi.

| | $T = 3$ | | | $T = 4$ | | |
|---|---|---|---|---|---|---|
| | SR↑ | mAcc↑ | mIoU↑ | SR↑ | mAcc↑ | mIoU↑ |
| w/o Viterbi | 27.48 | 56.62 | 68.92 | 14.85 | 51.46 | 66.47 |
| w/ Viterbi | **31.83** | **57.31** | **78.33** | **19.71** | **51.85** | **74.46** |

## C  FURTHER DISCUSSIONS

In this section, we provide further discussions with related works and insights that are not included in the main paper due to page limits.

**Limitations.** The limitations of our method are as follows. First, the model may fail to identify state changes if they are not explicitly shown in the video. This is a general challenge for procedure planning models, as the details of objects may be hard to recognize if they are far from the camera. Although we tried to tackle this challenge by associating non-visible state changes with language descriptions, which is learned from the paired vision-text training data, there is no guarantee that the non-visible state changes issue can be totally removed. A potential direction is to generalize the task by taking long-term videos as the input visual observations, so that the model can infer the state changes from video history or ASR narrations. Second, the quality of state descriptions relies on LLMs, which would be a bottleneck of state representation. We will explore more effective and reliable methods to leverage LLMs' commonsense knowledge, or utilize generative vision-language models (VLMs) for state description generation. Third, our approach is under a close-vocabulary setting. We will explore the open-vocabulary setting in the future.

**Differences from other mid-state prediction methods.** Differences from existing mid-state prediction methods: (1) Compared to PlaTe (Sun et al., 2022) and Ext-GAIL (Bi et al., 2021). First, PlaTe and Ext-GAIL are under the full supervision setting (i.e., visual observations of intermediate states are annotated, shown in Figure 1(b)) while ours is under the weak supervision setting (i.e., mid-state annotations are not available). Second, PlaTe formulates mid-state prediction as $P(s_t|s_{t-1}, a_{t-1}, s_T)$ conditioned on last state $s_{t-1}$, last action $a_{t-1}$, and end state $s_T$, Ext-GAIL formulates mid-state prediction as $P(s_t|s_{t-1}, a_{t-1}, z_c)$ where $z_c$ is context information learned from $s_0$ and $s_T$, while we formulate mid-state prediction as $P(s_{1:(T-1)}|s_0, s_T)$ conditioned on start state $s_0$ and end state $s_T$. Third, PlaTe represents mid-states using extremely low-dimensional (actually 4-d) features in their implementation, which is hard to learn and represent mid-state information. The role of their mid-state predictor is questionable.

(2) Compared to methods under the weak supervision settings. Recent works like P3IV (Zhao et al., 2022) consider the weak supervision where mid-states are not annotated, and they didn't predict mid-states in their pipeline. Differently, we leverage LLMs' commonsense knowledge to transform step into state change descriptions. Therefore, our work rethinks the role and representation style of mid-states, and we expect future works to further investigate the state representation.

**Comparisons with related works.** Some recent works explored how to extract commonsense information via text descriptions using language models. For example, Socratic Models (Zeng et al., 2022) (SMs) use language models in a zero-shot setting to solve downstream multimodal tasks. The similarity between ours and SMs is that we both leveraged LLMs and prompting to obtain commonsense knowledge for other modalities and multimodal tasks, especially video-related tasks and planning tasks. The main differences are: (1) Goal. SMs aim to embrace the heterogeneity of pretrained models through structured Socratic dialogue, while our goal is to represent steps as state changes via LLMs descriptions for procedure planning. (2) Task. SMs focus on zero-shot multimodal tasks without further training, while we focus on procedure learning in instructional videos with weak supervision and further training; (3) Framework. SMs deliver a zero-shot modular framework that composes multimodal models and LLMs and makes sample-specific API calling, while we only used LLMs once to obtain the generic language descriptions and train another planning model.

**Viterbi alrogithm.** We use Viterbi for post-processing during inference. We use transition matrix in Viterbi to include the temporal ordering knowledge in the training data, i.e., action co-occurrence frequencies. We empirically found that the training priors help with success rate increase. Our emission matrix estimation is a new contribution to deterministic modeling for procedure planning. Instead of sampling 1,500 generated sequences to estimate the emission matrix (Zhao et al., 2022), we run the feedforwarding only once and use the single predicted probability matrix as the emission matrix, which is simple and more time-efficient. This will be a useful tool for future deterministic modeling works.

**Hallucination of LLMs.** In the main paper, we observed the hallucination of LLMs using the baseline from Menon & Vondrick (2022), *e.g.*, given the action step "add onion" and the task goal "make kimchi fried rice", the generated description of the state before adding onion is "the onion was uncut and unchopped", which is incorrect because the onion should have been cut before being added to the rice. We proposed a chain-of-thought prompting method to first describe more details of the steps and then describe the state changes based on the detailed step descriptions. According to our manual checkup, most of the descriptions are reasonable and match human knowledge. We observed a few failure cases. One common problem is that LLMs may combine two steps as one. For example, for the step "cut strawberries" and task "make French strawberry cake", one of the after-state descriptions is "The strawberries are on the cake" which is incorrect. The reason is that LLMs mix two steps "cut strawberries" and "add strawberries" to generate the after-state descriptions. A potential solution is to require the LLMs to (1) distinguish different steps by feeding other steps into the prompt as references, (2) consider the temporal relations between steps.

**Generalization capabilities.** Our method can handle variations in action steps, object states, and environmental conditions. For variations in action steps, we evaluated our method with variant lengths of action steps, ranging from 3 to 6. According to the experimental results, our method consistently outperforms state-of-the-art models. For object states and environmental conditions, our evaluation benchmark instructional video datasets cover various topics and domains, including different environments like cooking, housework, and car repairing, and different objects like fruits, drinks, and household items. For example, CrossTask covers 133 types of steps in 18 tasks, and COIN covers 778 types of steps in 180 tasks. The evaluation on these two datasets can reflect the models' generality on various object states and environmental conditions.

**Potential benefit and drawback of mid-states.** The potential benefit of mid-states is the explainability, *e.g.*, extending language-only procedures to multi-modal procedures by adding intermediate visual states. The extension can be realized by (a) retrieving images or video clips from a predefined corpus of images or videos based on feature similarities; (b) generative images via text-to-image generation models (e.g., Stable Diffusion) based on the associated language descriptions. A potential drawback is the impact on the uncertainty modeling, *i.e.*, generating multiple procedures given the same start and goal states. As our motivation is to reduce the high variance of visual observations, language descriptions as the supervision of mid-states would also decrease the uncertainty of sequences. The future direction is to combine mid-states and language descriptions with probabilistic modeling.

**Noisy visual observations.** Visual scenes of states are diverse and may have low qualities (*i.e.*, incomplete or noisy) in instructional videos. Considering the high variance of visual observations, we proposed to represent visual states using LLMs descriptions and align the visual observations with language descriptions. We expect the discriminative language description to reduce the variance of visual states and add LLM commonsense knowledge for representation learning.

**Scaling up to larger dataset.** Different from existing benchmark datasets for procedure planning in instructional videos, larger datasets would have the following requirements: (1) Open-vocabulary setting. Large-scale real-word datasets are often open-vocabulary. For the state description generation process, since LLM is designed for open-vocabulary sequence generation, our generation part can be easily applied to larger datasets. For the step classifier, we can replace the fixed-vocabulary classifier (i.e., two-layer FFN) with a similarity calculation module, i.e., calculating the similarity between output embeddings of step decoder and language step descriptions. (2) Efficient training. Training on large-scale datasets often requires efficient training like distributed computing. Since our state decoder and step decoder are encoder-only Transformer architecture, it is easy to extend the training pipeline to distributed computing. (3) Long-term prediction. As shown in the comparison results with state-of-the-art methods, long-term prediction is still challenging for procedure planning model, but is an essential ability for larger datasets. One potential extension way is to treat our model as an adaptor to transform visual observations into tokens as the inputs to generative model like LLMs or VLMs for long-term sequence generation, which is a popular way to adapt LLMs or VLMs to downstream applications.

