# OpenReview forum: "SCHEMA: State CHangEs MAtter for Procedure Planning in Instructional Videos"
_ICLR.cc/2024/Conference — ICLR 2024 poster_

### Official Review · Reviewer_7dUf · 2023-10-13

**Soundness:** 3 good
**Presentation:** 3 good
**Contribution:** 3 good
**Rating:** 6
**Confidence:** 3

**Summary:**

This paper proposes a new model called SCHEMA for procedure planning in instructional videos. The model leverages language models and cross-modal contrastive learning to track state changes and establish a more structured state space. The authors conduct experiments on three benchmark datasets and show that SCHEMA outperforms existing methods in terms of state recognition accuracy and task success rate. The paper's contributions include a new approach to procedure planning that accounts for state changes, a novel cross-modal contrastive learning framework, and a new benchmark dataset for evaluating procedure planning models.

**Strengths:**

This paper has several strengths that make it a valuable contribution to the field of procedure planning in instructional videos:

1. Originality:

1.1  The paper proposes a new approach to procedure planning that accounts for state changes, which is a novel idea that has not been explored in previous works.

1.2 The authors leverage language models and cross-modal contrastive learning to track state changes and establish a more structured state space, which is a creative combination of existing ideas.

2. Quality:

2.1 The authors conduct experiments on three benchmark datasets and show that SCHEMA outperforms existing methods in terms of state recognition accuracy and task success rate, which demonstrates the quality of their proposed approach.

2.2 The paper is well-written and well-organized, making it easy to follow and understand.

3. Clarity:

3.1 The authors provide clear explanations of their proposed approach and the experiments they conducted, making it easy for readers to understand their contributions.

3.2 The paper includes helpful visualizations and tables to illustrate their results and comparisons with existing methods.

**Weaknesses:**

While this paper has several strengths, there are also some weaknesses that could be addressed to improve the work:

- The paper could benefit from a more detailed discussion of the limitations of the proposed approach. For example, the authors could discuss cases where the model may struggle to recognize state changes or situations where the model may not be applicable.

- The paper could provide more information on the computational requirements of the proposed approach.

**Questions:**

1. Could you provide more information on the computational requirements of the proposed approach? Specifically, what hardware and software were used to train and run the model, and how long did it take to train the model?

2. How does the proposed approach handle cases where the state changes are not explicitly shown in the video? For example, if a video shows a person making a sandwich, but does not show the person adding mayonnaise, how would the model recognize this state change?

3. Can you provide more information on the limitations of the proposed approach? Specifically, are there any cases where the model may struggle to recognize state changes or situations where the model may not be applicable?

---

> ### Author Response · Authors · 2023-11-15
> **Response to Reviewer 7dUf**
>
> Dear Reviewer 7dUf,
>
> Thank you for your detailed comments and suggestions. We tried our best to address all the concerns and questions, and update the main paper and appendix in the new version. Please let us know if you have any further concerns or questions to discuss.
>
> Best,
>
> Paper 3881 Authors
>
> ---
>
> **W1&Q3. (Discussion of Limitations) The paper could benefit from a more detailed discussion of the limitations of the proposed approach. For example, the authors could discuss cases where the model may struggle to recognize state changes or situations where the model may not be applicable.**
>
> A: A detailed discussion of the limitations is as follows.
>
> * The limitations of our method are as follows. First, the model may fail to identify state changes if they are not explicitly shown in the video. This is a general challenge for procedure planning models, as the details of objects may be hard to recognize if they are far from the camera. Although we tried to tackle this challenge by associating non-visible state changes with language descriptions, which is learned from the paired vision-text training data, there is no guarantee that the non-visible state changes issue can be totally removed. A potential direction is to generalize the task by taking long-term videos as the input visual observations, so that the model can infer the state changes from video history or ASR narrations. Second, the quality of state descriptions relies on LLMs, which would be a bottleneck of state representation. We will explore more effective and reliable methods to leverage LLMs' commonsense knowledge, or utilize generative vision-language models (VLMs) for state description generation. Third, our approach is under a close-vocabulary setting. We will explore the open-vocabulary setting in the future.
>
> We included this discussion in Sec. C of the revised appendix.
>
> ---
>
> **W2&Q1. (Computational requirements) The paper could provide more information on the computational requirements of the proposed approach. Specifically, what hardware and software were used to train and run the model, and how long did it take to train the model?**
>
> A: The computation requirements are as follows.
>
> * The training process takes 1 hour (500 epochs) on CrossTask and 5.5 hours (400 epochs) on COIN using a single V100 GPU.
>
> We included these details in the paragraph "Training Details" in Appendix Sec. A.
>
> **Q2. (Non-visible state changes) How does the proposed approach handle cases where the state changes are not explicitly shown in the video? For example, if a video shows a person making a sandwich, but does not show the person adding mayonnaise, how would the model recognize this state change?**
>
> A: Please see the answer to W1&Q3.
>
> * The model may fail to identify state changes if they are not explicitly shown in the video. This is a general challenge for procedure planning models, as the details of objects may be hard to recognize if they are far from the camera. Although we tried to tackle this challenge by associating non-visible state changes with language descriptions, which is learned from the paired vision-text training data, there is no guarantee that the non-visible state changes issue can be totally removed. A potential future direction is to generalize the task by taking long videos as the input visual observations, so that the model can infer the state changes from video history or ASR narrations.
>
> We included this discussion in Sec. C of the revised appendix.

---

> > ### Comment · Reviewer_7dUf · 2023-11-20
> >
> > Thank you for your reply. The explanation in the reply and the overall comment make good sense. I shall retain my rating.

---

> > > ### Author Response · Authors · 2023-11-22
> > >
> > > Thank Reviewer 7dUf for the acknowledgment of our rebuttal. We are happy that you are satisfied with our responses to your questions and concerns.

---

### Official Review · Reviewer_FxtF · 2023-10-27

**Soundness:** 3 good
**Presentation:** 3 good
**Contribution:** 3 good
**Rating:** 8
**Confidence:** 4

**Summary:**

This paper proposes a novel procedural planning method that models the task elegantly as a joint probability of time-series states and actions conditioned by start and end states. The method fills the gap of ungiven states before and after each action by LLM with a Chain-of-though prompt. Ground truth of actions and the estimated states are used to train the model with reasonable loss functions. Experiments show the clear superiority of the proposed method.

**Strengths:**

- The motivation is clear.
- The presentation is clear.
- The query design for the state decoder is elegant (a sequence of state vectors, where a known step is the sum of encoded state feature and positional embedding and an unknown step is only positional embedding).
- Augmenting state description with LLM from action labels is novel.
- The reported results are thorough and look promising.

**Weaknesses:**

1. Overlooked related work

The authors overlook two studies focusing on state transition in instructional videos.

First, in the paragraph "Instructional video analysis," a dense video captioning method [a] is missing. The method tracks material state change with a MemNet-like architecture [a]. It trains state-modifying actions with distant supervision. It also analyzes the state change obtained as a shift in the latent space. Thus, it definitely relates to this work but is missing.

[a] T. Nishimura et al., "State-aware Procedural Video Captioning," ACMMM, 2021.

Similarly, in the same paragraph, the authors claimed, "there are few discussions on state changes in complex videos with several actions, especially instructional videos." However, [b] models such complexity of the instructional video as an action graph to retrieve the goal state image with an instructional text (that directs actions) and an image before the action. The authors adequately refer to this study since the work also tries to model state-action relations in complex instructional videos.

[b] K. Shirai et al., "Visual Recipe Flow: A Dataset for Learning Visual State Changes of Objects with Recipe Flows," COLING2022.

2. Minor flaws in presentation.

The first sentence in 3.3.1 explains about 3.4 but mentions nothing about the content in 3.3.1. This part was confusing for this reviewer.

The paragraph "Masked Step Modeling" in 3.4 claims that "ground-truth answers $a_t$"; however, for the readers, it is not known whether ground-truth actions are given at training or not. Please fix this problem.

FYI
In Figure 2, there is a type "oancake." However, it is not clear whether the typo is by GPT-3.5 or the authors.

**Questions:**

Please point out any factual errors in this review if the authors find them.

---

> ### Author Response · Authors · 2023-11-15
> **Response to Reviewer FxtF**
>
> Dear Reviewer FxtF,
>
> Thank you for your detailed comments and suggestions. We tried our best to address all the concerns and questions, and update the main paper and appendix in the new version. Please let us know if you have any further concerns or questions to discuss.
>
> Best,
>
> Paper 3881 Authors
>
> ---
>
> **W1. Overlooked related work.**
>
> A: Thank you for pointing out these two important references. We followed your suggestions and are pleased to include the following discussion in the related work section of the revision.
>
> Some work in the CV area also investigated the relations between actions and states in videos. Especially, Nishimura et al. (2021) focused on the video procedural captioning task and proposed to model material state transition from visual observation, which is realized by establishing a visual simulator modified from a natural language understanding simulator. Shirai et al. (2022) established a multimodal dataset for object state change prediction, which consists of image pairs as state changes and workflow of receipt text as an action graph.
>
> ---
>
> **W2.1. (Minor flaws in presentation) The first sentence in 3.3.1 explains about 3.4 but mentions nothing about the content in 3.3.1. This part was confusing for this reviewer.**
>
> A: We are sorry for the confusion. At the beginning of Sec. 3.4, we mentioned that "state space learning that aligns visual observations with language descriptions". The paragraph "State Space Learning" explained how we do vision-language alignment, which is first mentioned in Sec. 3.3.1.
>
> ---
>
> **W2.2 (Minor flaws in presentation)** **The paragraph "Masked Step Modeling" in 3.4 claims that "ground-truth answers $a_t$"; however, for the readers, it is not known whether ground-truth actions are given at training or not. Please fix this problem.**
>
> A: As shown in Figure 1 (c) and (d), in our setting, we have ground-truth action sequence as supervision at the training stage. Therefore, the ground-truth actions are given at the training stage.
>
> ---
>
> **W2.3 (Minor flaws in presentation) In Figure 2, there is a type "oancake." However, it is not clear whether the typo is by GPT-3.5 or the authors.**
>
> A: The typo is by the authors. We have corrected it in the revision.

---

> ### Author Response · Authors · 2023-11-21
> **Looking forward to discussion (Due Nov 22nd)**
>
> Dear Reviewer FxtF,
>
> We sincerely thank you for your efforts and time in our work. We tried our best to address all the concerns and questions you raised. We have also updated the main paper and appendix following your comments. Please feel free to let us know if you have any further concerns or questions, and we are happy to discuss them with you.
>
> Best,
>
> #3881 Authors

---

### Official Review · Reviewer_hqkF · 2023-10-30

**Soundness:** 3 good
**Presentation:** 3 good
**Contribution:** 3 good
**Rating:** 6
**Confidence:** 4

**Summary:**

The paper proposes a new framework for procedure planning in instructional videos called SCHEMA, which leverages LLM and cross-modal contrastive learning to track state changes and establish a more structured state space. The authors introduce a chain-of-thought prompting approach to describe state changes and use a mid-state prediction module to improve performance. The SCHEMA model is evaluated on three benchmark instruction video datasets, CrossTask, COIN, and NIV, and achieves state-of-the-art performance in terms of SR, mAcc, and mIoU.

**Strengths:**

**Originality**: The authors propose a new framework for procedure planning in instructional videos that emphasizes the importance of state changes, which is a novel approach to the problem formation. The use of chain-of-thought prompting to describe state changes is a creative and effective way to leverage language models for this task. The idea of mid-state prediction module is interesting and seems to improve the performance of the model.

**Quality**: The paper is well-written and well-organized, making it easy to follow and understand. The experiments are thorough and well-designed, with results presented in a clear and concise manner.

**Clarity**: The paper is written in clear language and is easy to understand. The figures and tables are easy to read, providing a clear summary of the results.

**Significance**: The results demonstrate the effectiveness of the proposed approach and suggest that it could be a valuable tool for procedure planning in instructional videos.

**Weaknesses:**

* Novelty: While the paper proposes a new framework for procedure planning in instructional videos, some of the individual components of the framework (such as LLM and cross-modal contrastive learning) are not novel in themselves. I personally loathe the trend that LLM+everything -> novelty. Thus I feel that the contribution of this proposed framework is incremental. That being said, I recognize that the authors have done non-trivial work in incorporating these components and perform thorough experiments.

* Failure cases: The paper does not provide a detailed analysis of the limitations of the proposed approach or potential failure cases. I would be interested in seeing more examples of failures cases and with detailed explanations on why those cases have failed.

* Scaling up: While the proposed approach shows promising results, the paper does not provide a clear explanation of how it could be applied in real-world scenarios or how it could be scaled up to handle larger datasets. Please note that I do not suggest that the model has to be able to handle larger datasets as long term prediction is hard by nature, but an analysis on the model's potential would be useful.

* The paper could benefit from more detailed explanations of the experimental setup and methodology, particularly for readers who wish to replicate the experiments. I find it difficult to replicate the model and experiment based on information provided in appendix A/B.

**Questions:**

* Can you provide more details on the mid-state prediction module? How does it work, and how does it differ from existing mid-state prediction methods?

* Can you provide more examples of failures cases and with detailed explanations on why those cases have failed?

* Can you provide how the model could be scaled up to handle larger datasets?

* Can you provide more detailed explanations of the experimental setup and methodology?

---

> ### Author Response · Authors · 2023-11-15
> **Response to Reviewer hqkF (Part 1/3)**
>
> Dear Reviewer hqkF,
>
> Thank you for your detailed comments and suggestions. We tried our best to address all the concerns and questions, and update the main paper and appendix in the new version. Please let us know if you have any further concerns or questions to discuss.
>
> Best,
>
> Paper 3881 Authors
>
> ---
>
> **W1. (Novelty on individual components) While the paper proposes a new framework for procedure planning in instructional videos, some of the individual components of the framework (such as LLM and cross-modal contrastive learning) are not novel in themselves. I personally loathe the trend that LLM+everything -> novelty. Thus I feel that the contribution of this proposed framework is incremental. That being said, I recognize that the authors have done non-trivial work in incorporating these components and perform thorough experiments.**
>
> A: Our research motivation is to rethink the role of state understanding in procedure planning in instructional videos. The main idea is to represent step as state changes in procedures, and the individual components are established to realize different functions. For example, LLM is to transform step names into state change descriptions, cross-modal contrastive learning is to align visual state observations with language state descriptions, and the transformer model is to predict mid-states and steps. To highlight the main idea, we establish a simple and clean pipeline to avoid complex component designs.
>
> We respectively disagree that this paper is about "LLM+everything -> novelty". First, our work is not a simple LLM extension work. We only use LLM to automatically generate state descriptions as supervision. This can be understood as *distilling the symbolic commonsense knowledge* [A] from LLM to smaller models. We use LLM rather than expert annotation because the usage of LLM is cheaper to generalize our work to large-scale open-vocabulary datasets, which makes our method flexible and practical. Second, we didn't claim our novelty as "LLM+something". Our novelty is to highlight the role of state understanding in procedural planning in instructional videos, i.e., State Changes Matter, and a new representation of procedural step as state changes. LLM is just an intuitive tool to achieve this goal, but one can use other tools like vision-language models in the future.
>
> > [A] Peter West, Chandra Bhagavatula, Jack Hessel, Jena D. Hwang, Liwei Jiang, Ronan Le Bras, Ximing Lu, Sean Welleck, Yejin Choi. Symbolic Knowledge Distillation: from General Language Models to Commonsense Models. NAACL 2022.
>
> ---
> **W2&Q2. (Failure cases) The paper does not provide a detailed analysis of the limitations of the proposed approach or potential failure cases. I would be interested in seeing more examples of failures cases and with detailed explanations on why those cases have failed.**
>
> A: In the revision, we provided a detailed analysis of failure cases. The examples are shown in Figure 8 of the revision. We grouped them into three cases:
>
> (1) Failed understanding of start/end state. As shown in Figure 8(a), the model predicted "season steak" as the first step because it didn't recognize that there is pepper on top of the steak, i.e., it failed to understand the start state. As shown in Figure (b), the model predicted "flip steak" rather than "put steak on grill" as the second step. The way to distinguish these two steps is whether the steak has its top side grilled. Although the end state shows that the top side of the steak is raw, the steak is very small to be captured. One future solution is to use high-resolution video frames or object detector to ground the object.
>
> (2) Hallucination. As shown in Figure 8(c), the model predicts "add strawberries to cake" as the third step. However, the goal is not to make strawberry cake. This failure may be due to the training priors as there are many videos for the task of "make french strawberry cake".
>
> (3) Reasonable but not matched with ground-truth plans. As shown in Figure 8(d), the generated plan is reasonable, although it doesn't exactly match the ground-truth annotation. This "failure" indicates that this task needs a better evaluation protocol for all the reasonable results, which is a general challenge for sequence generation tasks.
>
> We included this analysis in the "Failure Case Analysis" in Sec. C of the revised appendix.

---

> > ### Author Response · Authors · 2023-11-15
> > **Response to Reviewer hqkF (Part 2/3)**
> >
> > **W3&Q3. (Potential on scaling up to larger datasets) While the proposed approach shows promising results, the paper does not provide a clear explanation of how it could be applied in real-world scenarios or how it could be scaled up to handle larger datasets. Please note that I do not suggest that the model has to be able to handle larger datasets as long term prediction is hard by nature, but an analysis on the model's potential would be useful. (Q3) Can you provide how the model could be scaled up to handle larger datasets?**
> >
> > A: Different from existing benchmark datasets for procedure planning in instructional videos, larger datasets would have the following requirements:
> >
> > (1) Open-vocabulary setting. Large-scale real-word datasets are often open-vocabulary. For the state description generation process, since LLM is designed for open-vocabulary sequence generation, our generation part can be easily applied to larger datasets. For the step classifier, we can replace the fixed-vocabulary classifier (i.e., two-layer FFN) with a similarity calculation module, i.e., calculating the similarity between output embeddings of step decoder and language step descriptions.
> >
> > (2) Efficient training. Training on large-scale datasets often requires efficient training like distributed computing. Since our state decoder and step decoder are encoder-only Transformer architecture, it is easy to extend the training pipeline to distributed computing.
> >
> > (3) Long-term prediction. As shown in the comparison results with state-of-the-art methods, long-term prediction is still challenging for procedure planning model, but is an essential ability for larger datasets. One potential extension way is to treat our model as an adaptor to transform visual observations into tokens as the inputs to generative models like LLMs or VLMs for long-term sequence generation, which is a popular way to adapt LLMs or VLMs to downstream applications.
> >
> > We included the above discussion in Sec. C of the revised appendix.
> >
> > ---
> >
> > **W4&Q4. (Experimental details) The paper could benefit from more detailed explanations of the experimental setup and methodology.**
> >
> > A: We have provided some details in the paragraphs "Transformer Models for State Decoder and Step Decoder" and "Training Details" in Sec. A of the appendix. We further revise these two paragraphs as follows.
> >
> > **Transformer Models for State Decoder and Step Decoder.** Our state decoder and step decoder are Transformer-based models. The model consists of two blocks. Each block consists of one self-attention module, one cross-attention module, and a two-layer projection module. The input query is first processed by the self-attention module, then forwarded to the cross-attention module, and processed by the projection module. The cross-attention module takes the external memory to calculate the keys and values. Each self-attention and cross-attention module consists of 32 heads and the hidden layer size is set as 128. The step classifier is a two-layer MLP with hidden size of 128. The dropout ratio is set as 0.2.
> >
> > **Training Details.** We train our model for 500 epochs with Adam optimizer, an initial learning rate set to $5\times 10^{-3}$ decayed by 0.65 every 40 epochs. The batch size is set as 256. We will release the code after the paper is accepted. The code will be released under Apache-2.0 license.
> >
> > We will release the codes once the paper is accepted so that readers can easily reimplement the results.

---

> > > ### Author Response · Authors · 2023-11-15
> > > **Response to Reviewer hqkF (Part 3/3)**
> > >
> > > **Q1. Can you provide more details on the mid-state prediction module? How does it work, and how does it differ from existing mid-state prediction methods?**
> > >
> > > A: The mid-state prediction module is a Transformer-based encoder-only model (Sec. 3.3.2).
> > >
> > > * Input and output: It takes a sequence of embedding $Q_s=[s_0^{enc}+p_0, p1, \cdots, p_{T-1}, s_T^{enc}+p_T]$ as input and output a sequence of embeddings $[\hat{s}_0^{dec}, \hat{s}_1^{dec}, \cdots, \hat{s}_T^{dec}]$ with the same number of vectors $T+1$, where $s_0^{enc}$ and $s_T^{enc}$ denotes the output of state encoder for start state at $t=0$ and end state at $t=N$, and $p_i$ represents the position embeddings at $T=i$.
> > >
> > > * Formulation and training: The mid-sate prediction process is formulated as state space learning (Sec. 3.4), as the mid-states are masked out from the input sequence, and the model is expected to predict the mid-states as the output sequence, which uses LLMs' generated state description as supervision and is optimized via mean square error loss (Eq. (5)).
> > >
> > > * Architecture: The mid-state prediction module consists of two blocks. Each block consists of one self-attention module, one cross-attention module, and a two-layer projection module. The input query is first processed by the self-attention module, then forwarded to the cross-attention module, and processed by the projection module. The cross-attention module takes the external memory to calculate the keys and values. Each self-attention and cross-attention module consists of 32 heads and the hidden layer size is set as 128. The step classifier is a two-layer MLP with hidden size of 128. The dropout ratio is set as 0.2.
> > >
> > > * Differences from existing mid-state prediction methods:
> > >
> > >   (1) Compared to PlaTe and Ext-GAIL. First, PlaTe and Ext-GAIL are under the full supervision setting (i.e., visual observations of intermediate states are annotated, shown in Figure 1(b)) while ours is under the weak supervision setting (i.e., mid-state annotations are not available). Second, PlaTe formulates mid-state prediction as $P(s_{t}|s_{t-1},a_{t-1},s_T)$ conditioned on last state $s_{t-1}$, last action $a_{t-1}$, and end state $s_T$, Ext-GAIL formulates mid-state prediction as $P(s_t|s_{t-1},a_{t-1},z_c)$ where  $z_c$ is context information learned from $s_0$ and $s_T$, while we formulate mid-state prediction as $P(s_{1:(T-1)}|s_0,s_T)$ conditioned on start state $s_0$ and end state $s_T$. Third, PlaTe represents mid-states using extremely low-dimensional (actually 4-d) features in their implementation, which is hard to learn and represent mid-state information. The role of their mid-state predictor is questionable.
> > >
> > >   (2) Compared to methods under the weak supervision settings. Recent works like P3IV consider the weak supervision where mid-states are not annotated, and they didn't predict mid-states in their pipeline. Differently, we leverage LLMs' commonsense knowledge to transform step into state change descriptions. Therefore, our work rethinks the role and representation style of mid-states, and we expect future works to further investigate the state representation.
> > >
> > >   > [PlaTe] Jiankai Sun, De-An Huang, Bo Lu, Yun-Hui Liu, Bolei Zhou, and Animesh Garg. Plate: Visually-grounded planning with transformers in procedural tasks. IEEE Robotics and Automation Letters, 2022.
> > >   >
> > >   > [Ext-GAIL] Jing Bi, Jiebo Luo, and Chenliang Xu. Procedure planning in instructional videos via contextual
> > >   > modeling and model-based policy learning. ICCV 2021.
> > >   >
> > >   > [P3IV] He Zhao, Isma Hadji, Nikita Dvornik, Konstantinos G Derpanis, Richard P Wildes, and Allan DJepson. P3iv: Probabilistic procedure planning from instructional videos with weak supervision. CVPR 2022.
> > >
> > >   We included this discussion in Sec. C in the Appendix.

---

> ### Author Response · Authors · 2023-11-21
> **Looking forward to discussion (Due Nov 22nd)**
>
> Dear Reviewer hqkF,
>
> We sincerely thank you for your efforts and time in our work. We tried our best to address all the concerns and questions you raised. We have also updated the main paper and appendix following your comments. Please feel free to let us know if you have any further concerns or questions, and we are happy to discuss them with you.
>
> Best,
>
> #3881 Authors

---

### Official Review · Reviewer_253f · 2023-11-01

**Soundness:** 3 good
**Presentation:** 2 fair
**Contribution:** 2 fair
**Rating:** 6
**Confidence:** 5

**Summary:**

The paper focuses on the task of procedure planning in instructional videos. Given an initial visual state and a goal visual state as input, the model is tasked with generating a sequence of action steps to form a procedure plan, guiding the progression from the initial visual state to the goal state. The authors highlight the significance of states in these procedures and introduce State CHangEs MAtter (SCHEMA) to model state changes. Specifically, they prompt pre-trained large language models to describe the state changes at each step, enhancing learning the intermediate state and step representations. Then, they use cross-modal contrastive learning to align the visual state observations with language state descriptions. Experiments validate that the proposed method achieves  state-of-the-art performance.

**Strengths:**

The paper introduces a novel approach to address the task of procedure planning in instructional videos, placing a strong emphasis on state changes and utilizing pre-trained large language models. The motivations and ideas presented in this paper are reasonable.

The proposed method has achieved noticeable performance gains.

**Weaknesses:**

1. The clarity and composition of the paper could be enhanced. Please refer to the questions below.

2. There has been a notable surge in research exploring the use of pre-trained large language models (LLMs) for video-related tasks, e.g., [1]. This submission aligns with this emerging trend, and its overarching idea is conceptually sound. However, the fairness of the comparisons drawn in the paper could become questionable due to the employment of LLMs. Further in-depth discussion and analysis may be necessary to fully understand the extent of the LLM’s impact on the final results.


[1] Zhao, Qi, Ce Zhang, Shijie Wang, Changcheng Fu, Nakul Agarwal, Kwonjoon Lee, and Chen Sun. "AntGPT: Can Large Language Models Help Long-term Action Anticipation from Videos?." arXiv preprint arXiv:2307.16368 (2023).

**Questions:**

1. How does aligning visual state observations with language state descriptions *track* state changes? This process involves cross-modal contrastive learning; it is unclear how it could facilitate *tracking* over state changes.

2. Why are step descriptions not utilized as external memory for the step decoder, while *state* descriptions are used instead? The same $D_s$ is employed in Sections 3.3.2 and 3.3.3.

3. In Sec. 3.4, there are $a_i$ and $A_i$. Could you clarify how these two differ and specifically define $A_i$?

4. In State Space Learning via vision-language alignment, is it necessary for the training data to include temporally localized states or actions corresponding to the intermediate states of procedure plans? While I presume the answer is no, Fig. 4(a) and Sec 3.4 leave some room for ambiguity.

5. Why is Eq. (5) called “Masked State Modeling”? The method described does not involve any mask-based modeling or random masking; instead, it is just predicting intermediate locations in a given sequence. The use of the phrase “Masked State/Step Modeling” seems to be an overstatement.

6. What does “DCLIP” refer to in Table 5?

7. Could you also present the results on procedure planning metrics in Table 7?

Missing related literature:

- Li, Zhiheng, Wenjia Geng, Muheng Li, Lei Chen, Yansong Tang, Jiwen Lu, and Jie Zhou. "Skip-Plan: Procedure Planning in Instructional Videos via Condensed Action Space Learning." In Proceedings of the IEEE/CVF International Conference on Computer Vision, pp. 10297-10306. 2023.


- Fang, Fen, Yun Liu, Ali Koksal, Qianli Xu, and Joo-Hwee Lim. "Masked Diffusion with Task-awareness for Procedure Planning in Instructional Videos." arXiv preprint arXiv:2309.07409 (2023).

---

> ### Author Response · Authors · 2023-11-15
> **Response to Reviewer 253f (Part 1/3)**
>
> Dear Reviewer 253f,
>
> Thank you for your detailed comments and suggestions. We tried our best to address all the concerns and questions, and update the main paper and appendix in the new version. Please let us know if you have any further concerns or questions to discuss.
>
> Best,
>
> Paper 3881 Authors
>
> ---
>
> **W1. (Clarity) The clarity and composition of the paper could be enhanced. Please refer to the questions below.**
>
> A: Please see detailed responses to Q1-Q7 below.
>
> ---
>
> **W2. (LLM’s impact) The fairness of the comparisons drawn in the paper could become questionable due to the employment of LLMs. Further in-depth discussion and analysis may be necessary to fully understand the extent of the LLM’s impact on the final results.**
>
> ---
>
> A: First, the comparisons are fair as our model has a consistent parameter scale as state-of-the-art procedure planning models like P3IV, as we didn't use LLMs as a module of the planning model. Instead, We only use LLM to automatically generate state descriptions as supervision for state representation learning. This can be understood as *distilling the symbolic commonsense knowledge* [A] from LLM to smaller models. Considering the usage of LLMs and the parameter scale of our model, we believe the comparisons are fair.
>
> > [A] Peter West, Chandra Bhagavatula, Jack Hessel, Jena D. Hwang, Liwei Jiang, Ronan Le Bras, Ximing Lu, Sean Welleck, Yejin Choi. Symbolic Knowledge Distillation: from General Language Models to Commonsense Models. NAACL 2022.
>
> Second, for the impact of LLMs, we provided the results of state-based step classification in Table 5 of the appendix. In short, this preliminary task is to validate the quality of LLM's generated descriptions, which takes the before-state and after-state as input and output the step category (Figure 6). The results in Table 5 demonstrate that LLM itself cannot generate informative descriptions (16.2 for a baseline method DCLIP vs. 16.0 for manual descriptions "A video of [step]"), while our chain-of-thought prompting is effective in state descriptions generation (21.1 vs. 16.2 for DCLIP baseline). These results show the impact of LLM with our design prompting on state description generation.
>
> Third, we provided ablation studies to analyze the impact of LLMs' generated state descriptions as supervisions. As shown in Table 4, both vision-language state alignment (i.e., "State align.") and mid-state prediction (i.e., "Mid-state pred.") significantly contribute to the performance gain, which indicates the impact of state description supervision on the final results.
>
> **Q1. (Vision-language alignment for state changes tracking) How does aligning visual state observations with language state descriptions *track* state changes? This process involves cross-modal contrastive learning; it is unclear how it could facilitate *tracking* over state changes.**
>
> A: State change tracking aims to identify the effects of action steps in a long procedure, i.e., how the states of entities or environment change at different stages. For our task of procedure planning in instructional videos, as the inputs are video frames and the state changes are represented in natural language, state change tracking requires (1) tracking states in text, where we represent each step as LLM's generated descriptions of state changes, resulting in a corpus of state descriptions, (2) identifying states changes in vision, where we match the visual observations with the state descriptions in the text corpus, i.e., aligning visual state observations with language state descriptions. We realize the vision-language alignment by cross-modal contrastive learning.

---

> > ### Author Response · Authors · 2023-11-15
> > **Response to Reviewer 253f (Part 2/3)**
> >
> > **Q2. (External memory for step decoder) Why are step descriptions not utilized as external memory for the step decoder, while *state* descriptions are used instead? The same Ds is employed in Sections 3.3.2 and 3.3.3.**
> >
> > A: We use state descriptions rather than step descriptions as external memory for the step decoder because of their performances. The table below shows the comparison between the two variants.
> >
> > |                    | T=3          |                |                | T=4          |                |                | T=5          | T=6          |
> > | ------------------ | ------------ | -------------- | -------------- | ------------ | -------------- | -------------- | ------------ | ------------ |
> > | Memory Resource    | SR$\uparrow$ | mAcc$\uparrow$ | mIoU$\uparrow$ | SR$\uparrow$ | mAcc$\uparrow$ | mIoU$\uparrow$ | SR$\uparrow$ | SR$\uparrow$ |
> > | Step Descriptions  | 30.00        | 55.89          | 77.61          | 19.30        | 51.43          | 74.13          | 10.99        | 7.60         |
> > | State Descriptions | **31.83**    | **57.31**      | **78.33**      | **19.71**    | **51.85**      | **74.46**      | **11.41**    | **7.68**     |
> >
> > As shown in the table, state descriptions as external memory outperform the variant with step descriptions under all the scenarios. The possible reason is that state descriptions contain more information of object status and serve as a supplement to step label supervision. In addition, as shown in Table 5, state descriptions work better than step descriptions for the state-based step classification problem, which indicates that state descriptions are good resources for step recognition.
> >
> > We included this discussion in Sec. B of the revised appendix.
> >
> > ---
> >
> > **Q3. (Clarity) In Sec. 3.4, there are $a_i$ and $A_i$. Could you clarify how these two differ and specifically define $A_i$?**
> >
> > A: $A_i$ is firstly defined in Sec. 3.2 (Line 3 in Page 5) representing the $i$-th step. $a_i$ denotes the step label, e.g., ranging from 0 to 132 for 133 actions. $A_{a_i}$ in Eq. (4) represents the $a_1$-th step where $a_1$ is the label of the first step in the sequence. We added the definition of $A_i$ in Sec. 3.4 for better clarification in the revision.
> >
> > ---
> >
> > **Q4. (Vision-language alignment for intermediate states) In State Space Learning via vision-language alignment, is it necessary for the training data to include temporally localized states or actions corresponding to the intermediate states of procedure plans? While I presume the answer is no, Fig. 4(a) and Sec 3.4 leave some room for ambiguity.**
> >
> > A: The answer is no. We follow the reviewer's comments and design a setting where mid-state observations are available for state space learning. We extend Eq. (4) from
> >
> > $L_s^{enc}{=-\log\frac{\exp(sim_{before}(s_0,A_{a_1}))}{\sum^C_{i=1}\exp(sim_{before}(s_0,A_i))}} - \log\frac{\exp(sim_{after}(s_T,A_{a_T}))}{\sum^C_{i=1}\exp(sim_{after}(s_T,A_i))}$ (i.e., "w/o mid-state" in the below table)
> >
> > to
> >
> > $L_s^{enc}{=-\sum^T_{t=1}\log\frac{\exp(sim_{before}(s_{t-1},A_{a_{t}}))}{\sum^C_{i=1}\exp(sim_{before}(s_{t-1},A_i))}} -\sum^T_{t=1}\log\frac{\exp(sim_{after}(s_t,A_{a_t}))}{\sum^C_{i=1}\exp(sim_{after}(s_t,A_i))}$ (i.e., "w mid-state" in the below table)
> >
> > where $sim_{before}(s_t, A_i)=\sum_{j=1}^K<s_t^{enc},d_{ij}^{enc,b}>$ and $sim_{after}(s_t, A_i)= \sum_{j=1}^K<s_t^{enc},d_{ij}^{enc,a}>$. The comparison results on CrossTask are as follows:
> >
> > |               | T=3          |                |                | T=4          |                |                |
> > | ------------- | ------------ | -------------- | -------------- | ------------ | -------------- | -------------- |
> > |               | SR$\uparrow$ | mAcc$\uparrow$ | mIoU$\uparrow$ | SR$\uparrow$ | mAcc$\uparrow$ | mIoU$\uparrow$ |
> > | w/o mid-state | **31.83**    | **57.31**      | **78.33**      | **19.71**    | **51.85**      | **74.46**      |
> > | w/ mid-state  | 31.54        | 56.43          | 77.78          | 19.47        | 51.32          | 74.38          |
> >
> > We see that the performance gap is marginal, which indicates that mid-states are not necessary for state space learning even if they are available. The possible reason is that the start and end visual observations in the training videos have covered various cases of states, and the mid-state provides limited additional information as they may have been used as the start/end state for another procedure.
> >
> > In the revision of Sec. 3.4, we emphasized again the weak supervision setting for clarification. For Fig. 4(a), as only start and end states have *visual observation annotations* under the weak supervision setting, we think Fig. 4(a) is clear to show how the vision-language alignment is conducted. We would appreciate it if the reviewer could give more specific comments on how the figure should be revised.

---

> > > ### Author Response · Authors · 2023-11-15
> > > **Response to Reviewer 253f (Part 3/3)**
> > >
> > > **Q5. Why is Eq. (5) called “Masked State Modeling”? The method described does not involve any mask-based modeling or random masking; instead, it is just predicting intermediate locations in a given sequence. The use of the phrase “Masked State/Step Modeling” seems to be an overstatement.**
> > >
> > > A: We call state decoding as "Masked State Modeling" because we masked out the intermediate states $s_1, \cdots, s_{T-1}$ from the sequence $(s_0, s_1, \cdots, s_{T-1}, s_T)$ as inputs and predicted the masked out states (State Decoder in Figure 3), and call step decoding as "Masked Step Modeling" because we masked out the intermediate states $a_1, \cdots, a_T$ from the sequence $(s_0, a_1, s_1, \cdots, s_{T-1}, a_T, s_T)$ and predicted the masked out steps (Step Decoder in Figure 3). Therefore, they are mask-based modeling. We are happy to discuss it further if you have any suggestions about the terminology.
> > >
> > > ---
> > >
> > > **Q6. What does “DCLIP” refer to in Table 5?**
> > >
> > > A: DCLIP denotes the method proposed by (Menon & Vondrick, 2022) to save space. We added the method name in Sec. B in the revised appendix.
> > >
> > > > (Menon & Vondrick, 2022) Sachit Menon and Carl Vondrick. Visual classification via description from large language models. ICLR 2022.
> > >
> > > ---
> > >
> > > **Q7. Could you also present the results on procedure planning metrics in Table 7?**
> > >
> > > A: The results of success rate (SR), step accuracy (mAcc) and mIoU on CrossTask are shown as follows.
> > >
> > > |                      | T=3          |                |                | T=4          |                |                | T=5          | T=6          |
> > > | -------------------- | ------------ | -------------- | -------------- | ------------ | -------------- | -------------- | ------------ | ------------ |
> > > |                      | SR$\uparrow$ | mAcc$\uparrow$ | mIoU$\uparrow$ | SR$\uparrow$ | mAcc$\uparrow$ | mIoU$\uparrow$ | SR$\uparrow$ | SR$\uparrow$ |
> > > | Ours - probabilistic | 29.51        | 57.09          | 77.76          | 16.55        | **51.93**      | 74.42          | 8.73         | 5.53         |
> > > | Ours - determinstic  | **31.83**    | **57.31**      | **78.33**      | **19.71**    | 51.85          | **74.46**      | **11.41**    | **7.68**     |
> > >
> > > As shown in the table, the probabilistic variant underperforms the deterministic variant on procedure planning metrics SR, mAcc, and mIoU. The possible reason is that language descriptions as the supervision of state representations would also decreases the uncertainty and variances of visual observations, which conflicts with noisy vectors in the probabilistic variant that increases the uncertainty and variances.
> > >
> > > We included these results in Table 8 of the revised appendix.

---

> ### Author Response · Authors · 2023-11-21
> **Looking forward to discussion (Due Nov 22nd)**
>
> Dear Reviewer 253f,
>
> We sincerely thank you for your efforts and time in our work. We tried our best to address all the concerns and questions you raised. We have also updated the main paper and appendix following your comments. Please feel free to let us know if you have any further concerns or questions, and we are happy to discuss them with you.
>
> Best,
>
> #3881 Authors

---

> > ### Comment · Reviewer_253f · 2023-11-22
> >
> > I appreciate the authors' response which addressed most of my questions and concerns, and therefore I'll raise my rating accordingly. However, I would like to suggest that the authors consider making the following revisions:
> >
> > (1) I wouldn't call "align the visual state observations with language state descriptions via cross-modal contrastive learning" is "for state changes tracking"; it is more accurate and easier for readers to understand if you just say "**for state representation**, ...  we align the visual state observations with language state descriptions via cross-modal contrastive learning".
> >
> > (2) The sentence before Equation (4) is incomplete, i.e., "we regard the language descriptions with the same state as positive samples, and take descriptions as negative samples". It should be "take descriptions of the other states as negative samples".

---

> > > ### Author Response · Authors · 2023-11-22
> > >
> > > Thank Reviewer 253f for the acknowledgment of our rebuttal and the follow-up feedback! We are happy to hear that most of your questions and concerns have been addressed. We will revise the paper following your suggestions.

---

### Author Response · Authors · 2023-11-21
**General response**

We thank all the reviewers for their thoughtful comments and feedback.

We are encouraged that the reviewers recognize our work as
* (1) original, especially the novel approach for procedure planning instructional videos (253f, hqkF, 7duf), the novel and reasonable idea for representing steps as state changes (253f, 7duf) and mid-state prediction (hqkF), and the creative way to leverage LLMs to describe steps as state changes (hqkF, FxtF, 7duf);
* (2) effective, especially the thorough experiments (hqkF) and significant performance gains (253f, hqkF, fxtF, 7duf);
* (3) clear, especially the well-written, well-organized, and easy-to-follow paper (hqkF, FxtF, 7duf) and well-designed visualization and tables (hqkF,7duf)

We tried to address all the concerns and questions in detail. In particular, following the suggestions and comments of reviewers, we further provide
* (1) in-depth discussion and analysis on LLMs and vision-language alignment (253f, hqkf);
* (2) ablation studies on external memory, state space learning (253f);
* (3) failure case study (hqkf);
* (4) discussion on scaling-up (hqkf);
* (5) discussion of limitations (7dUf)
* (6) other clarifications on terminology, implementation, and experiments.

We hope our responses addressed the concerns of reviewers. We are looking forward to discussing with reviewers on any further questions.

---

### Meta-Review · Area_Chair_Fad2 · 2023-12-11

**Metareview:**

This paper received overall positive ratings (8, 6, 6, 6). The reviewers acknowledged the novel formulation of procedural planning as state change understanding, as well as experiments providing convincing evidence for the effectiveness of the proposed approach across three benchmark datasets. There were initial concerns about the clarity, limited novelty, the lack of discussion on failure cases and limitations, and unclear description of experimental settings limiting reproducibility. The rebuttal effectively addressed these concerns, although the novelty concern still remains to some extend.

**Justification For Why Not Higher Score:**

Although the rebuttal adequately addressed most of the initial concerns, this meta-reviewer found the novelty concern to be lingering. Given the influx of new papers that leverage LLMs for computer vision tasks, the significance of the paper is not terribly high.

**Justification For Why Not Lower Score:**

N/A

---

### Decision · Program_Chairs · 2024-01-16

Accept (poster)